# Mapping the planet's critical natural assets

Rebecca Chaplin-Kramer [1,2,3] ✉, Rachel A. Neugarten [4,5], Richard P. Sharp[1], Pamela M. Collins[5], Stephen Polasky [6], David Hole [5], Richard Schuster[7,8], Matthew Strimas-Mackey [9], Mark Mulligan[10], Carter Brandon[11], Sandra Diaz [12,13], Etienne Fluet-Chouinard [14], L. J. Gorenflo [15], Justin A. Johnson[6], Christina M. Kennedy [16], Patrick W. Keys [17], Kate Longley-Wood [18], Peter B. McIntyre[4], Monica Noon[5], Unai Pascual [19,20,21], Catherine Reidy Liermann[22], Patrick R. Roehrdanz [5], Guido Schmidt-Traub[23], M. Rebecca Shaw[24], Mark Spalding[18,25], Will R. Turner [5], Arnout van Soesbergen [10,26] & Reg A. Watson [27]

Sustaining the organisms, ecosystems and processes that underpin human wellbeing is necessary to achieve sustainable development. Here we define critical natural assets as the natural and semi-natural ecosystems that provide 90% of the total current magnitude of 14 types of nature's contributions to people (NCP), and we map the global locations of these critical natural assets at 2 km resolution. Critical natural assets for maintaining local-scale NCP (12 of the 14 NCP) account for 30% of total global land area and 24% of national territorial waters, while 44% of land area is required to also maintain two global-scale NCP (carbon storage and moisture recycling). These areas overlap substantially with cultural diversity (areas containing 96% of global languages) and biodiversity (covering area requirements for 73% of birds and 66% of mammals). At least 87% of the world's population live in the areas benefitting from critical natural assets for local-scale NCP, while only 16% live on the lands containing these assets. Many of the NCP mapped here are left out of international agreements focused on conserving species or mitigating climate change, yet this analysis shows that explicitly prioritizing critical natural assets and the NCP they provide could simultaneously advance development, climate and conservation goals.

Human actions are rapidly transforming the planet, driving losses of nature at an unprecedented rate that negatively impacts societies and economies, from accelerating climate change to increasing zoonotic pandemic risk[1,2]. Recognizing the accelerating severity of the environmental crisis, the global community committed to Sustainable Development Goals and the Paris Agreement on climate change in 2015. In 2022, the UN Convention on Biological Diversity (CBD) will adopt new targets for conserving, restoring and sustainably managing multiple dimensions of biodiversity, including nature's contributions to people (NCP)[3]. Collectively, these three policy frameworks will shape the sustainable development agenda for the next decade. All three depend

heavily on safeguarding natural assets[1,4], the living components of our lands and waters. For instance, restoring and ending conversion and degradation of forests, wetlands and peatlands could sequester 9 Gt $CO_2$ per year by 2050 (ref. [5]). While ambitious new targets to protect species and ecosystems have been proposed, including 'half Earth' (conserving half the Earth's area for nature)[6] and '30 by 30' (30% protected by 2030) (ref. [7]), these targets have been criticized for insufficiently accounting for the needs of people, including many Indigenous and local communities[8]. It is therefore essential to demonstrate how nature conservation contributes to human wellbeing. Yet despite the urgency of safeguarding natural assets around the world, we still have limited

understanding of the spatial extent of ecosystems providing essential benefits to humanity[9].

In this Article, leveraging recent advances in scientific understanding, data availability (Supplementary Tables 1 and 2) and computational power, we undertake a global analysis of 14 NCP (Extended Data Fig. 1), the most comprehensive set mapped globally so far[10,11]. Most of these NCP are considered ecosystem services, which, according to the IPBES Conceptual Framework[3,12], are embedded within the larger, more inclusive concept of NCP, and therefore we use the latter term throughout to better align with international frameworks and policy[13]. Twelve of the NCP included in our analysis deliver primarily local benefits (though some subsequently enter global supply chains), including contributions to the provision of food, energy and raw materials; the regulation of water quality and disaster risk; and recreational activities (Fig. 1a). We prioritize these 12 'local' NCP at the country level to identify areas needed to maintain their consistent provision within each country. In contrast, we prioritize at the global scale for two NCP related to climate regulation (terrestrial ecosystem carbon storage and vegetation-regulated atmospheric moisture recycling), whose benefits accrue at continental scales or globally.

Through multi-objective spatial optimization at a resolution of 2 km, we map the location of the planet's critical natural assets, defined as the natural and semi-natural ecosystems providing 90% of current levels of each NCP (that is, locations required for all NCP to meet or exceed a 90% target). Beyond this target there are diminishing returns, with disproportionately more natural area required to reach incrementally higher magnitudes of provision of NCP (Fig. 1b). We use the term 'critical' to convey that losing these natural assets would lead to disproportionately large losses in NCP. This usage is related to, though not the same as, the term 'critical natural capital', which has been formally defined in the ecological economics literature as ecosystems providing important functions that cannot be substituted[14]. However, as we do not explore substitutes to natural capital found in alternative forms of capital (that is, manufactured capital), we use the term 'natural assets' instead to avoid confusion. The assets we focus on are natural and semi-natural ecosystems (Supplementary Table 3), excluding developed lands (croplands and urban areas) and unvegetated areas, to provide insights for conservation priorities relevant to the CBD; global priorities for restoration[15] or for management of developed areas[16] are beyond the scope of this first effort to map critical natural assets. Our analysis reveals three key findings about critical natural assets: (1) their extent and location; (2) the number of people benefitting from, and living within, these critical areas; and (3) the degree of overlap between critical natural assets delivering local benefits and those delivering global benefits and between critical natural assets and other global priorities for the CBD (biodiversity and cultural diversity).

## Results and discussion

### Extent and location of critical natural assets

Critical natural assets providing the 12 local NCP (Fig. 1a) occupy only 30% (41 million km²) of total land area (excluding Antarctica) and 24% (34 million km²) of marine Exclusive Economic Zones (EEZs), reflecting the steep slope of the aggregate NCP accumulation curve (Fig. 1b). Despite this modest proportion of global land area, the shares of countries' land areas that are designated as critical can vary substantially. The 20 largest countries require only 24% of their land area, on average, to maintain 90% of current levels of NCP, while smaller countries (10,000 to 1.5 million km²) require on average 40% of their land area (Supplementary Data 1). This high variability in the NCP–area relationship is primarily driven by the proportion of countries' land areas made up by natural assets (that is, excluding barren, ice and snow, and developed lands), but even when this is accounted for, there are outliers (Extended Data Fig. 2). Outliers may be due to spatial patterns in human population density (for example, countries with dense population centres and vast expanses with few people, such

as Canada and Russia, require far less area to achieve NCP targets) or large ecosystem heterogeneity (if greater ecosystem diversity yields higher levels of diverse NCP in a smaller proportion of area, which may explain patterns in Chile and Australia).

The highest-value critical natural assets (the locations delivering the highest magnitudes of NCP in the smallest area, denoted by the darkest blue or green shades in Fig. 1c) often coincide with diverse, relatively intact natural areas near or upstream from large numbers of people. Many of these high-value areas coincide with areas of greatest spatial congruence among multiple NCP (Extended Data Fig. 3). Spatially correlated pairs of local NCP (Supplementary Table 4) include those related to water (flood risk reduction with nitrogen retention and nitrogen with sediment retention); forest products (timber and fuelwood); and those occurring closer to human-modified habitats (pollination with nature access and with nitrogen retention). Coastal risk reduction, forage production for grazing, and riverine fish harvest are the most spatially distinct from other local NCP. In the marine realm, there is substantial overlap of fisheries with coastal risk reduction and reef tourism (though not between the latter two, which each have much smaller critical areas than exist for fisheries).

### Number of people benefitting from critical natural assets

We estimate that ~87% of the world's current population, 6.4 billion people, benefit directly from at least one of the 12 local NCP provided by critical natural assets, while only 16% live on the lands providing these benefits (and they may also benefit; Fig. 2a). To quantify the number of beneficiaries of critical natural assets, we spatially delineate their benefitting areas (which varies on the basis of NCP: for example, areas downstream, within the floodplain, in low-lying areas near the coast, or accessible by a short travel). While our optimization selects for the provision of 90% of the current value of each NCP, it is not guaranteed that 90% of the world's population would benefit (since it does not include considerations for redundancy in adjacent pixels and therefore many of the areas selected benefit the same populations), so it is notable that an estimated 87% do. This estimate of 'local' beneficiaries probably underestimates the total number of people benefitting because it includes only NCP for which beneficiaries can be spatially delineated to avoid double-counting, yet it is striking that the vast majority, 6.1 billion people, live within 1 h travel (by road, rail, boat or foot, taking the fastest path[17]) of critical natural assets, and more than half of the world's population lives downstream of these areas (Fig. 2b). Material NCP are often delivered locally, but many also enter global supply chains, making it difficult to delineate beneficiaries spatially for these NCP. However, past studies have calculated that globally more than 54 million people benefit directly from the timber industry[18], 157 million from riverine fisheries[19], 565 million from marine fisheries[20] and 1.3 billion from livestock grazing[21], and across the tropics alone 2.7 billion are estimated to be dependent on nature for one or more basic needs[22].

Nearly all countries have a large percentage (>80%) of their populations benefitting from critical natural assets, but small countries have much larger proportions of their populations living within the footprint of critical natural assets than do large countries (Fig. 2a and Supplementary Data 2). When people live in these areas, and especially when current levels of use of natural assets are not sustainable, regulations or incentives may be needed to maintain the benefits these assets provide. While protected areas are an important conservation strategy, they represent only 15% of the critical natural assets for local NCP (Supplementary Table 5); additional areas should not necessarily be protected using designations that restrict human access and use, or they could cease to provide some of the diverse values that make them so critical[23]. Other area-based conservation measures, such as those based on Indigenous and local communities' governance systems, Payments for Ecosystem Services programmes, and sustainable use of land- and seascapes, can all contribute to maintaining critical flows of NCP in natural and semi-natural ecosystems[24].

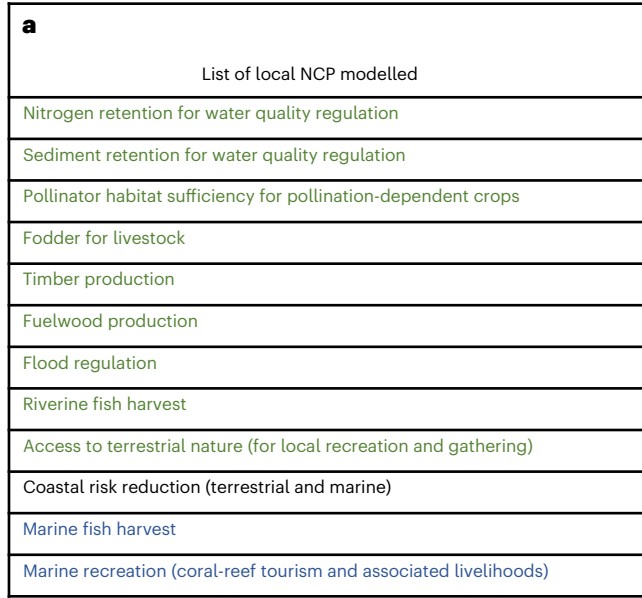

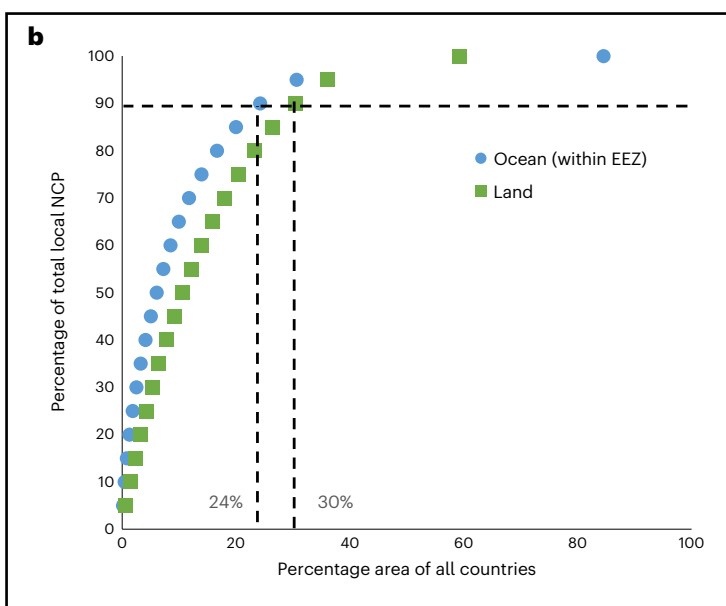

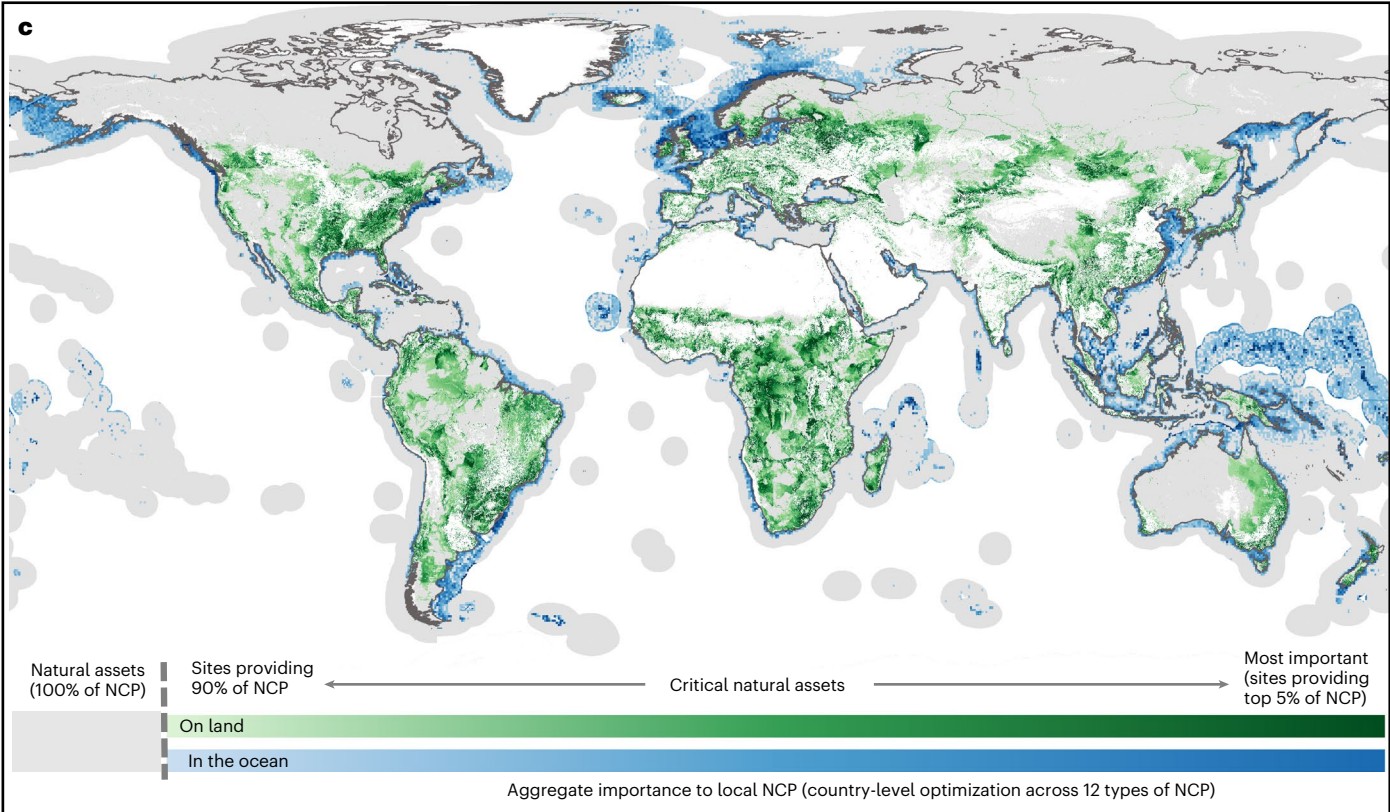

**Fig. 1 | Critical natural assets, defined as the natural and semi-natural terrestrial and aquatic ecosystems required to maintain 12 of nature's 'local' contributions to people (local NCP) on land (green) and in the ocean (blue). a**, The 12 local NCP analysed (that is, not including global NCP, shown in Supplementary Fig. 4). **b**, The NCP accumulation curve, reflecting the total area required to maintain target levels of all NCP in every country, with dotted lines denoting the area of critical natural assets (90% of NCP in 30% of land area and 24% of EEZ area). Areas selected by optimization within each country are aggregated across all countries to create a single global accumulation curve; for area requirements in individual countries, see Supplementary Data 1. **c**, Map of critical natural assets, with darker shades connoting critical natural assets that are associated with higher levels of aggregated NCP. Grey areas show the extent of remaining natural assets not designated 'critical' but included in this analysis; white areas (cropland, urban and bare areas, ice and snow, and ocean areas outside the EEZ) were excluded from the optimization.

## Overlaps between local and global priorities

Unlike the 12 local NCP prioritized here at the national scale, certain benefits of natural assets accrue continentally or even globally. We therefore optimize two additional NCP at a global scale: vulnerable terrestrial ecosystem carbon storage (that is, the amount of total ecosystem carbon lost in a typical disturbance event[25], hereafter 'ecosystem carbon') and vegetation-regulated atmospheric moisture recycling (the supply of atmospheric moisture and precipitation sustained by plant life[26], hereafter 'moisture recycling'). Over 80% of the natural asset locations identified as critical for the 12 local NCP are also critical

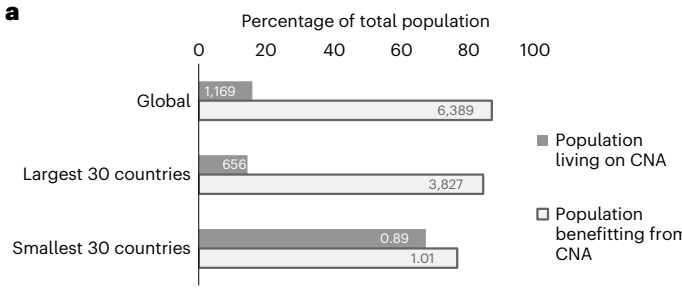

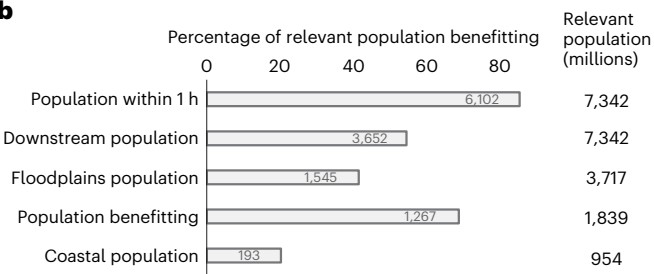

**Fig. 2 | People benefitting from and living on critical natural assets (CNA). a,b,** 'Local' beneficiaries were calculated through the intersection of areas benefitting from different NCP, to avoid double-counting people in areas of overlap; only those NCP for which beneficiaries could be spatially delineated were included (that is, not material NCP that enter global supply chains: fisheries, timber, livestock or crop pollination). Bars show percentages of total population globally and for large and small countries (**a**) or the percentage of relevant population globally (**b**). Numbers inset in bars show millions of people making up that percentage. Numbers to the right of bars in **b** show total relevant population (in millions of people, equivalent to total global population from Landscan 2017 for population within 1 h travel or downstream, but limited to the total population living within 10 km of floodplains or along coastlines <10 m above mean sea level for floodplain and coastal populations protected, respectively, and to rural poor populations for fuelwood).

for the two global NCP (Fig. 3). The spatial overlap between critical natural assets for local and global NCP accounts for 24% of land area, with an additional 14% of land area critical for global NCP that is not considered critical for local NCP (Extended Data Fig. 4). Together, critical natural assets for securing both local and global NCP require 44% of total global land area. When each NCP is optimized individually (carbon and moisture NCP at the global scale; the other 12 at the country scale), the overlap between carbon or moisture NCP and the other NCP exceeds 50% for all terrestrial (and freshwater) NCP except coastal risk reduction (which overlaps only 36% with ecosystem carbon, 5% with moisture recycling; Supplementary Table 4).

Synergies can also be found between NCP and biodiversity and cultural diversity. Critical natural assets for local NCP at national levels overlap with part or all of the area of habitat (AOH, mapped on the basis of species range maps, habitat preferences and elevation[27]) for 60% of 28,177 terrestrial vertebrates (Supplementary Data 3). Birds (73%) and mammals (66%) are better represented than reptiles and amphibians (44%). However, these critical natural assets represent only 34% of the area for endemic vertebrate species (with 100% of their AOH located within a given country; Supplementary Data 3) and 16% of the area for all vertebrates if using a more conservative representation target framework based on the IUCN Red List criteria (though, notably, achieving Red List representation targets is impossible for 24% of species without restoration or other expansion of existing AOH; Supplementary Data 4). Cultural diversity (proxied by linguistic diversity) has far higher overlaps with critical natural assets than does biodiversity; these areas intersect 96% of global Indigenous and non-migrant languages[28] (Supplementary Data 5). The degree to which languages are represented in association with critical natural assets is consistent across most

countries, even at the high end of language diversity (countries containing >100 Indigenous and non-migrant languages, such as Indonesia, Nigeria and India). This high correspondence provides further support for the importance of safeguarding rights to access critical natural assets, especially for Indigenous cultures that benefit from and help maintain them. Despite the larger land area required for maintaining the global NCP compared with local NCP, global NCP priority areas overlap with slightly fewer languages (92%) and with only 2% more species (60% of species AOH), although a substantially greater overlap is seen with global NCP if Red List criteria are considered (36% compared with 16% for local NCP; Supplementary Data 4). These results provide different insights than previous efforts at smaller scales, particularly a similar exercise in Europe that found less overlap with priority areas for biodiversity and NCP[29]. However, the 40% of all vertebrate species whose habitats did not overlap with critical natural assets could drive very different patterns if biodiversity were included in the optimization.

Although these 14 NCP are not comprehensive of the myriad ways that nature benefits and is valued by people[23], they capture, spatially and thematically, many elements explicitly mentioned in the First Draft of the CBD's post-2020 Global Biodiversity Framework[13]: food security, water security, protection from hazards and extreme events, livelihoods and access to green and blue spaces. Our emphasis here is to highlight the contributions of natural and semi-natural ecosystems to human wellbeing, specifically contributions that are often overlooked in mainstream conservation and development policies around the world. For example, considerations for global food security often include only crop production rather than nature's contributions to it via pollination or vegetation-mediated precipitation, or livestock production without partitioning out the contribution of grasslands from more intensified feed production.

## Gaps and next steps

Our synthesis of these 14 NCP represents a substantial advance beyond other global prioritizations that include NCP limited to ecosystem carbon stocks, fresh water and marine fisheries[30–32], though still falls short of including all important contributions of nature such as its relational values[33]. Despite the omission of many NCP that were not able to be mapped, further analyses indicate that results are fairly robust to inclusion of additional NCP. Dropping one of the 12 local NCP at a time results in <1–3% change in the total global land area required to maintain 90% of current levels of these NCP (Supplementary Table 6) and a high degree of spatial agreement. In fact, 62% of the total area on land is shared by all optimization solutions, and 97% of the area is included in 11 of the 12 solutions; similar values are found across most countries (Supplementary Data 1). Nevertheless, this same multi-NCP optimization approach could accommodate additional NCP as spatial data become available at sufficient resolution and appropriate scales.

There also is uncertainty in the identification of critical natural assets related to model error in the individual NCP that we were able to include. We acknowledge that NCP models, like all models, have errors, and that consistent global-scale modelling will miss details important for certain specific locations. Validation of NCP is particularly difficult given that there are no direct measurements for many NCP with assessment reliant on remotely sensed proxies. We utilize the best available global modelling approaches and data, most of which have been validated in at least some locations[19,25,34–41]. Where uncertainty existed about what distance was most appropriate to model the delivery of NCP (for example, how far to model people downstream or how far people might travel to natural assets), we performed further sensitivity analysis and confirmed that the estimated land area of critical natural assets is robust to the distance chosen (impacting results by <5%; Supplementary Table 6). As availability of global models for many of these NCP increases, future work should move towards ensemble modelling approaches, which have been shown to increase accuracy and reduce uncertainty compared with individual model outputs[42,43].

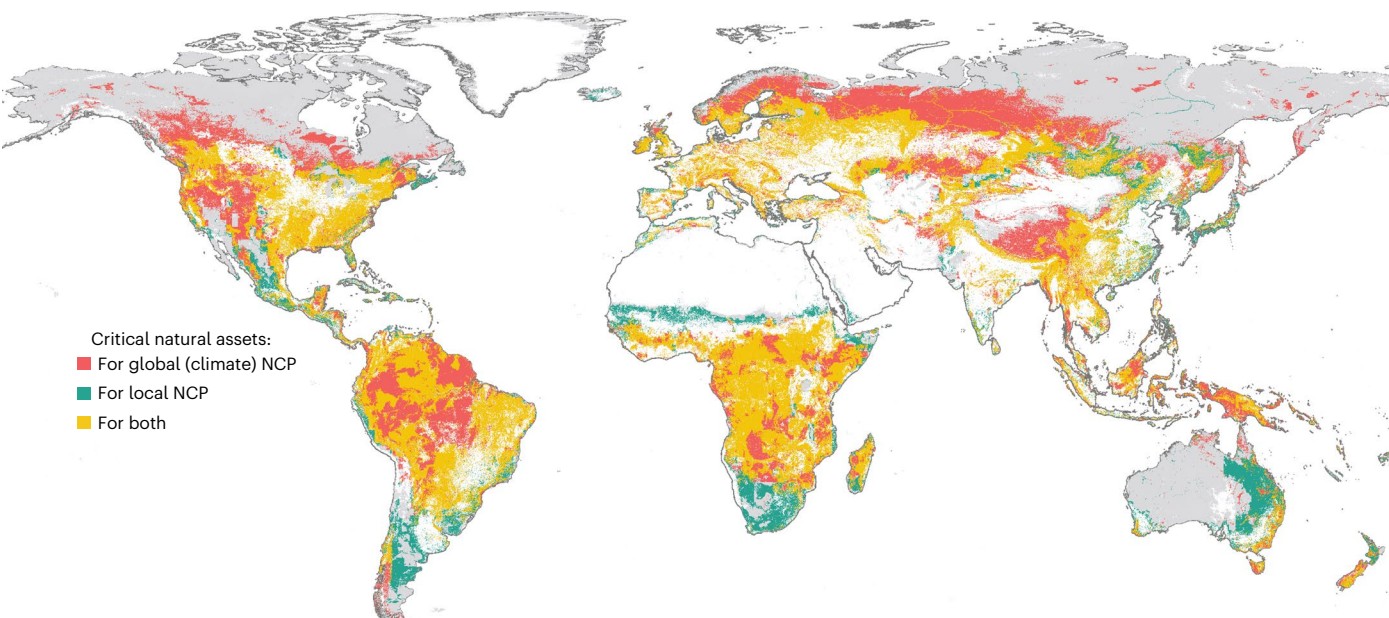

**Fig. 3 | Spatial overlaps between critical natural assets for local and global NCP.** Red and teal denote where critical natural assets for global NCP (providing 90% of ecosystem carbon and moisture recycling globally) or for local NCP (providing 90% of the 12 NCP listed in Fig. 1), respectively, but not both, occur; gold shows areas where the two overlap (24% of the total area). Together, local and global critical natural assets account for 44% of total global land area (excluding Antarctica). Grey areas show natural assets not defined as 'critical' by this analysis, though still providing some values to certain populations. White areas were excluded from the optimization.

**Critical natural assets:**
- For global (climate) NCP
- For local NCP
- For both

Data and modelling gaps prevented a broader exploration of issues relevant to the ecological supply side of NCP. Although results presented here suggest that nationally prioritized areas for local NCP can deliver on global priorities in many regions, they also highlight a need for integrated modelling to represent interactions between different NCP. For example, atmospheric moisture evapotranspired by Amazonian forests falls as rain in other parts of South America, supporting ecosystems that provide food, fuel and other benefits[26]. Further work is needed to move beyond the spatial overlaps explored here towards understanding functional inter-dependencies between NCP. We also acknowledge that urban and cropland systems were omitted from this analysis owing to data and modelling limitations that would fail to adequately capture the NCP supported by different land use types and land management practices within those systems. Likewise, arctic and desert ecosystems, owing to sparse vegetation and low human population densities, are not well represented in our NCP models and yet are very important to the people who live in and depend on them. As data and modelling gaps are filled, future assessment of critical natural assets should expand to recognize unique contributions of currently undervalued ecosystems and should include possible gains from restoring and sustainably managing human-dominated systems[15,16] to consider how these different conservation strategies can complement one another.

There are also, perhaps even more pronounced, data and modelling gaps to fill on the social side of NCP. In the NCP modelled here, we represent realized benefits of natural assets—weighted by beneficiary population when feasible—but this understates the range of ways in which natural assets directly and indirectly contribute to people's wellbeing. Limited socio-economic data and lack of reliable models linking NCP to wellbeing indicators preclude more precise valuation of most NCP at the global scale. Additional insight could be gained from mapping critical natural assets that support the most vulnerable or dependent[22] people, including Indigenous peoples whose livelihoods and cultural identities directly depend on nature (and indeed overlap substantially with critical natural assets, on the basis of our estimates of Indigenous language diversity on these lands; Supplementary Data 5), and the poor who may lack access to anthropogenic substitutes for NCP (see also philosophical considerations in Supplementary Note 1). Recent progress in linking ecological modelling with integrated assessment modelling and general equilibrium economic modelling[44] shows great promise for assessing the benefits of critical natural assets to society and the global economy. Such efforts could also reveal telecoupling of critical natural assets arising from transboundary flows between countries such as via international trade[45].

Finally, further work is needed to investigate whether critical natural assets are necessary or sufficient for meeting humanity's needs, by considering the availability of substitutes and what constitutes 'enough'. Though motivated by the ecological economic concept of 'critical natural capital', we were not able to capture the degree to which anthropogenic assets could replace natural assets. However, given the high correspondence of many NCP (Supplementary Table 4), it seems unlikely that anthropogenic assets could substitute for all benefits provided by natural assets in a particular area (for example, green infrastructure delivers many co-benefits beyond the single benefit built infrastructure is designed to deliver[46,47]). It seems more likely that more than 44% of the planet may be required to secure the 14 NCP mapped here (let alone the other diverse values of nature we were not able to map), most importantly because it is unclear how much of the current need for nature is already unmet. In many parts of the world, natural ecosystems are already degraded or converted, so maintaining 90% of current levels of NCP may be far too little (for example, places prone to catastrophic flooding due to habitat conversion[48], where grazing lands have been desertified[49] and where fish populations have crashed[50]). Furthermore, what is critical now may change in the future owing to climate change, population growth or change in technology or consumption patterns[1,11]. Other areas may not directly benefit people but may be critical to avoid ecological tipping points and collapse of NCP in affected ecosystems[51,52]. We therefore propose the analysis conducted here as a first attempt to define a minimum set of critical natural assets, and suggest that the overall approach provides a useful framework for exploring such issues with future scenario modelling to develop more resilient conservation for nature and people.

We acknowledge that our approach is strongly anthropocentric (NCP, and all the concepts included in them, are by definition anthropocentric[3]). As such, it is not intended to capture the intrinsic values of nature, or the value of the ecosystems or species providing the NCP highlighted here to other non-human organisms. Our focus and findings should not be interpreted as dismissing those values, and further work could explore natural assets providing important contributions to biodiversity that are not captured by species maps alone. For example, regulating contributions, including water quality regulation, natural hazards resilience, pollination and atmospheric moisture recycling, maintain the conditions under which current biodiversity thrives. Delineating species or high-biodiversity areas as the 'beneficiaries' for many of these contributions may be an important step towards reflecting nature's contributions to nature.

## Conclusion

Identifying critical natural assets could enable national and global leaders to prioritize the conservation of a wide range of NCP. We find it encouraging that securing 90% of the NCP mapped here is feasible with an area comparable to other proposed conservation targets[6,7,32,53]. Global analyses such as this can set a broader context for local decisions including understanding of distant connections that extend outside a country's borders, provide rapid information for global actors such as CBD signatories and non-governmental organizations working on conservation priorities across many countries, and supplement gaps in local information while it is still being generated[54]. National- and local-level use of spatial data is timely and relevant for countries seeking to access scarce international Sustainable Development Goal-related finance, as it can improve the ecological, social and economic aspects of project readiness[55,56]. However, we emphasize the value of the approach developed here more than the maps or data. This approach can be adapted and refined at the scales at which policy implementation occurs, with the best available data and complemented by input from local experts and diverse stakeholders, to improve accuracy and public legitimacy[57,58] and to ensure that human rights and diverse human relationships with nature are safeguarded. Moreover, creation and use of spatially explicit information allows for a focus on ecosystem quality over quantity, helping to avoid potentially perverse outcomes of area targets for conservation[59]. This approach for identifying critical natural assets is a vital step forward in empowering actors at all levels to make decisions that benefit both nature and people.

## Methods

### Modelling NCP

The 14 NCP in this analysis (Extended Data Fig. 1) were chosen to span development and climate goals, and to be mappable with spatially explicit data representing the period 2000–2020. We use European Space Agency (ESA) 2015, for land cover at 10 arcsec (~300 m at the equator) resolution, Landscan 2017 for population[60] at 30 arcsec (~1 km) resolution (which were the most current data available at the time we began our analysis). We focus on 'nature's contributions' to key benefits of interest (for example, security in food, water, hazards, materials and culture), meaning we partition out the role of natural and semi-natural ecosystems in producing those benefits. For food security, we include the contributions of pollinator habitat sufficiency to pollination-dependent crop production, vegetation-mediated atmospheric moisture recycling to crop and livestock production (included as a global NCP that also links to climate security), grassland fodder production to livestock production, and wild riverine and marine fisheries. For water security, we include the contributions of water quality regulation, via sediment retention and nutrient retention, but not water yield since the role of ecosystems in determining the quantity of water is minimal (other than by evapotranspiration, which is already captured in the vegetation-mediated moisture recycling, and regulation of timing of flows which is captured in flood risk reduction). For security of

protection from natural hazards, we include flood risk reduction and coastal risk reduction. For materials, we include timber production, fuelwood production and access to nature (which could be used for gathering, and also links to culture). For cultural benefits we include coral reef tourism (as the only globally mapped form of marine-based tourism) and access to nature again (which in addition to gathering also captures recreation or other uses of nearby greenspace). Finally, for climate security, in addition to moisture recycling mentioned above, we include total ecosystem carbon storage (as a global NCP). Below we briefly summarize the models used to map these local NCP (Supplementary Table 1) and global NCP (Supplementary Table 2); full information on each model is available in Supplementary Note 1.

### Local NCP.

1. Nitrogen retention to regulate water quality for downstream populations is modelled at native ESA resolution (at 10 arcsec or 300 m) using the InVEST[61] Nutrient Delivery Ratio model, which is based on fertilizer application, precipitation, topography and the retention capacity of vegetation, and has been previously used in global applications[11]. The people benefitting from nitrogen retention are those who would otherwise be exposed to nitrogen contamination in their drinking water. In this analysis, the number of people downstream was calculated for every pixel of habitat, to provide a sense of which habitat potentially benefits the most people. Ideally, to map realized nitrogen retention, we would be able to convert biophysical service production into a measure of change in wellbeing, whether monetary, in health terms or otherwise. However, the state of the science and data available globally precludes this for most services, so our proxy was the number of people downstream who could potentially benefit from that retention. NCP for nitrogen retention is expressed as nitrogen retention on natural and semi-natural pixels multiplied by the number of people downstream of those pixels (for more details, see Supplementary Note 1 Section 1).

2. Sediment retention to regulate water quality for downstream populations is modelled at native ESA resolution (10 arcsec or 300 m) by adapting the InVEST Sediment Delivery Ratio model, which maps overland sediment generation and delivery to the stream using the Revised Universal Soil Loss Equation (RUSLE) based on climate, soil texture, topography and land cover, and a conductivity index based on the upslope and downslope areas of each pixel. Ideally sediment retention would be delineated for reservoirs, irrigation canals or other water delivery infrastructure that is most impacted by sedimentation, but lacking a comprehensive global dataset identifying all such infrastructure, we again use the proxy of number of people downstream (as described for nitrogen retention, above). NCP for sediment retention is expressed as sediment retention on natural and semi-natural pixels multiplied by the number of people downstream of those pixels (Supplementary Note 1 Section 2).

3. Pollinator habitat sufficiency for pollination-dependent crop production is modelled at native ESA resolution (10 arcsec or 300 m) with a simplified version of InVEST, improving upon previous global mapping of the potential contribution of wild pollinators to nutrition production by mapping the value from the farm field back to habitat[11]. This contribution is calculated as the sufficiency of habitat surrounding farmland multiplied by the pollination dependency of crops (the proportion of yields that are pollination dependent, based on the crop mixes grown there). While this does not capture other important features that affect pollinators, such as presence of specific species important for nesting or floral resources or the intensity or frequency of pesticide applications, it provides a proxy of the

contribution of nearby natural habitat to pollination. This NCP is expressed in terms of the average equivalent number of people fed by pollination-dependent crop production, attributed to nearby ecosystems based on the area of pollinator habitat within pollinator flight distance of crops (Supplementary Note 1 Section 3).

4. Fodder production for livestock is modelled using version 3 of Co\$ting Nature[62] at a resolution of 5 arcmin (~10 km), and then masked to relevant ESA natural and semi-natural habitats (at 10 arcsec or 300 m resolution; Supplementary Table 3). Supply of fodder is calculated from remotely sensed dry matter productivity for the non-cropland cover fraction, and demand is estimated by the headcount of livestock in a grid cell multiplied by the average biomass intake requirements per animal. NCP for fodder production for livestock is expressed in terms of an index (0–1), re-scaled from the realized service, which is reported as the smaller of the supply or demand (if consumption exceeds productivity, the gap is assumed to be met with feed). The best available global inputs for dry matter productivity, livestock headcount, cropland and land cover are used as input (Supplementary Note 1 Section 4).

5. Timber production includes commercial (for example, for trade or export) and domestic (for example, for local use) timber, modelled using version 3 of Co\$ting Nature as two spatially mutually exclusive layers, because they represent two different sets of beneficiaries. This model outputs data at a resolution of 5 arcmin (~10 km), and is then masked to relevant ESA natural and semi-natural habitats (at 10 arcsec or 300 m resolution; Supplementary Table 3). NCP for timber production is expressed as an index (0–1) based on remotely sensed forest productivity and accessibility for harvest. Total potential sustainable supply of timber is estimated from the best available global aboveground carbon stock map multiplied by fractional tree cover for rural areas only. The sustainable harvest is calculated as the reciprocal of the number of years taken to develop the stock at the annual sequestration rate, according to dry matter productivity data. Demand is calculated differently for commercial versus domestic timber based on different assumptions of accessibility. Commercial accessibility is defined as within 6 h travel time of a population centre of >50,000 people and on slope gradients <70%. Domestic accessibility is defined as areas inaccessible for commercial harvest, and harvest rates are based on per capita consumption multiplied by population within 10 km (Supplementary Note 1 Section 5).

6. Fuelwood production is calculated as a byproduct of the timber model from version 3 of Co\$ting Nature. NCP for fuelwood production, like timber, is represented as an index (0–1) based on forest productivity and accessibility for harvest, but in this case specifically by rural people. Fuelwood can overlap spatially with domestic and commercial timber use, given that domestic and commercial timber harvest will not consume all sustainably available woody biomass in all places, owing to the slope gradient limit and/or in places where demand is less than supply, and fuelwood is often a byproduct of timber harvest. This model outputs data at a resolution of 5 arcmin (~10 km), and is then masked to relevant ESA natural and semi-natural habitats (at 10 arcsec or 300 m resolution; Supplementary Table 3) (Supplementary Note 1 Section 6).

7. Flood regulation is modelled using version 2 of the hydrologic model WaterWorld[35]. To map nature's influence on flood risk reduction, we identify the upstream places where canopies, wetlands and soils (green storage) retain and slowly release rainfall, to the benefit of downstream communities. NCP for flood regulation is expressed as an index (0–1) of 'green' water storage (based on wetland storage, canopy storage and soil

storage) multiplied by the number of people downstream. This model outputs data at a resolution of 5 arcmin (~10 km), and is then masked to ESA natural and semi-natural habitats (at 10 arcsec or 300 m resolution) (Supplementary Note 1 Section 7).

8. Access to nature is used as a proxy for numerous direct and indirect benefits to people, such as recreation, hunting and gathering, aesthetics, mental and physical health, and other cultural values that depend on the ability of people to access nature. This proxy NCP is expressed as the number of urban and rural[63] people within 1 h (or 6 h, for sensitivity analysis) travel of natural and semi-natural habitat, taking the least-cost path (by foot, road, rail or boat) across a friction layer[17] at a resolution of 2 km (equal-area projection) (Supplementary Note 1 Section 8).

9. Riverine fish catch is based on spatial disaggregation of nationally reported catch for 2007–2014 (ref. [19]) and updated to include catch estimated by household consumption surveys in 32 countries with severe underreporting[64]. Catches from large lakes were excluded. To spatially disaggregate the global catch of $13.3 \times 10^6$ tonnes within country borders, a multiple linear regression model of total fish catch in river basins compiled from the literature was fitted with three predictor variables: population density, river discharge and percentage wetland cover ($n = 40$, $R^2_{adj} = 0.69$). NCP for riverine fish harvest is represented as metric tonnes of fish caught per km[2] of land area per year, spatially allocated to the locations of the harvest. This model outputs data at a resolution of 5 arcmin (~10 km), and is then masked to ESA natural and semi-natural habitats (at 10 arcsec or 300 m resolution) (Supplementary Note 1 Section 9).

10. Marine fish catch is based on data from the Sea Around Us, Global Fishing Watch, and other sources to map fish catch for 2010–2014 within 30 min (~55 km) grid cells across the ocean[65,66]. NCP for marine fish harvest is represented as metric tonnes of fish caught per km[2] per year, spatially disaggregated to the locations of the catch (Supplementary Note 1 Section 10).

11. Coral reef tourism is taken from the Mapping Ocean Wealth dataset[34], which reports NCP for coral-reef-associated tourism as dollars of tourism expenditure (in deciles 1–10) at 15 arcsec (~500 m) resolution. National expenditure data are spatially distributed on the basis of three independent sources: hotel rooms from the commercial Global Accommodation Reference Database, dive shops and dive sites from Diveboard, and user-generated photos from the image-sharing website Flickr (Supplementary Note 1 Section 11).

12. Coastal risk reduction is modelled with InVEST for terrestrial and coastal/off-shore habitats[67–70] at native ESA resolution (10 arcsec or 300 m) updating previous global modelling[11] through the inclusion of new data and projecting the value from the shoreline back to the protective habitat (which may be off-shore). Coastal risk reduction depends on the physical exposure to coastal hazards (based on wind, waves, sea level rise, geomorphology and bathymetry), with and without natural habitat to attenuate storm surge, and the people exposed. NCP for coastal risk reduction is expressed as a unitless index of the coastal risk reduced by habitat multiplied by the number of people within the protective distance of the habitat (Supplementary Note 1 Section 12).

**Global NCP.**

1. Vulnerable terrestrial ecosystem carbon storage is mapped at a resolution of 1 arcsec (~30 m), as the aboveground and belowground ecosystem carbon lost in a 'typical' disturbance event, rather than the total stock[25]. This includes terrestrial and coastal (mangrove, salt marsh and seagrass) ecosystem carbon pools (aboveground, belowground and soils), based on what

carbon is likely to be released if the ecosystem were converted (Supplementary Note 1 Section 13).

2. Atmospheric moisture recycling is the process of water arising from the surface of the Earth as evaporation, flowing through the atmosphere as water vapour, and returning to the surface of the Earth as precipitation. Sources of evaporation include canopy interception, soil interception, soil evaporation, vegetation transpiration and open water evaporation. We employ an Eulerian moisture tracking model, WAM-2 layers[26], to quantify the source of moisture, where it travels through the atmosphere and where it falls out downwind across a 1.5° grid (~166 km). NCP of moisture recycling, which is to say the moisture associated with vegetation, is expressed as the volume of water evapotranspired that falls on all rainfed productive land (cropland, rangelands and working forests[71]) (Supplementary Note 1 Section 14).

### Attribution of value to natural assets

The first step in identifying critical natural assets, before the optimization can select the highest value areas across all NCP, is to attribute the magnitude of benefits and, where possible, the number of beneficiaries, to the ecosystems providing those benefits for each individual NCP on the basis of its unique characteristics (for example, attributing the value of pollination occurring on croplands to the nearby habitat supplying the pollinators, or the coastal risk reduced and number of people protected along the coastline to off-shore as well as on-shore habitats). We define natural assets as natural and semi-natural terrestrial ecosystems (including semi-natural lands like rangelands and production forests, but excluding cropland, urban areas, bare areas and permanent snow and ice; Supplementary Table 3) and inland and marine waters. Model outputs for pollination and coastal risk reduction are mapped back to habitat on the basis of pollinator flight distance (Supplementary Note 1 Section 3) and protective distance of coastal habitat (Supplementary Note 1 Section 12), respectively. For sediment and nitrogen retention, the count of people downstream of each habitat pixel was summed according to a hydrologic flow accumulation (Supplementary Note 1 Section 1), and for nature access, the count of people was calculated for each pixels by summing the population pixels within 1 h travel time according to a friction surface (Supplementary Note 1 Section 8). All other model outputs are coarser than ESA resolution and are masked to the land cover types defined as natural assets relevant to that NCP (for example, only forests for timber but excluding forests for grazing; Supplementary Table 3).

### Optimization of NCP

Using integer linear programming (prioritizr, Supplementary Note 1 Section 22), we identify minimum areas required (1) within each country's land borders and marine EEZs for the local NCP and (2) within all global land area (excluding Antarctica) and all countries' combined EEZ area for the global NCP, to reach target levels (ranging from 5% to 100%) of every NCP. Before optimization the data were re-sampled to a 2 km equal-area projection for consistency (Supplementary Note 1 Section 24). The optimization selects for the highest values across all NCP, providing the most benefit and/or to the greatest number of people, but not accounting for complementarity or redundancies of adjacent pixels (that is, not dynamically optimizing after each pixel's selection). Land and marine borders are based on Flanders Marine Institute (2020; Supplementary Table 2), and overlapping claims are excluded from the national analyses. The 12 local NCP are optimized for each country, then aggregated globally, while the two global NCP are optimized globally. In addition to these two main optimizations, we assess the sensitivity to scale by optimizing the 12 local NCP globally (instead of by each country; Extended Data Fig. 5), both with and without the two global NCP, and by substituting different scales of

beneficiaries mapping for people downstream and for access to nature (Supplementary Table 6). We also assess the sensitivity of the area and location of critical natural assets (the optimization solution for the 90% target) to different NCP combinations. These variations include optimizing for each NCP individually, and optimizing for all NCP but dropping each local NCP from the set of 12 to evaluate its effect on the overall optimization (Supplementary Table 6). We also examine the correspondence between NCP and the robustness of these different solutions, by calculating the percentage of area shared by different pairs of services (Supplementary Table 4) or the percentage of area shared by all solutions (Supplementary Data 1). We summarize the land and ocean areas required by country in Supplementary Data 1.

### Number of people benefitting from critical natural assets

We map the areas benefitting from critical natural assets to calculate the number of direct beneficiaries of these assets, and to compare the number of beneficiaries to the number of people living on the lands comprising these critical natural assets. For this analysis we are only able to include NCP for which the flow of the benefit can be spatially delineated: downstream water quality regulation (sediment retention and nitrogen retention), flood mitigation, nature access, fuelwood provision and coastal risk reduction. The benefitting areas of some of the material NCP that are traded (fish, timber, livestock and crops that are pollinated) or the location of people who buy those traded goods are not easily mapped, so people benefitting from these NCP are not included in this analysis of beneficiaries. However, people within 1 h of critical natural assets may provide a surrogate for many of the material NCP that are locally consumed. For water quality regulation, we take the population within the areas downstream (Supplementary Note 1 Section 1) of critical natural assets. For nature access, we take the population within 1 h travel (by foot, car, boat or rail; Supplementary Note 1 Section 8) of critical natural assets. Likewise, for the other NCP we take the relevant population downstream, within the protective distance, or a gathering distance of critical natural assets. The relevant population for each NCP is considered to be the total global population for nature access and water quality regulation, but is limited to the total population living within 10 km of floodplains for flood mitigation, population along coastlines (in exposed areas: <10 m above mean sea level) for coastal risk reduction, and rural poor populations for fuelwood. Total 'local' beneficiaries are calculated through the intersection of areas benefitting from different NCP, to avoid double-counting people in areas of overlap. We calculate the number of people and percent of relevant population benefitting globally for each NCP (Fig. 2b) and the total 'local' beneficiaries globally (Fig. 2a) and by country (Supplementary Data 2).

### Overlap analysis

We evaluate how well local and global critical natural assets align spatially with each other, and with biodiversity (terrestrial vertebrate species AOH[27]; Supplementary Note 1 Section 15) and cultural diversity (proxied by the number of Indigenous and non-migrant languages[28]; Supplementary Note 1 Section 16), to identify synergies between these different potential priorities. To examine the level of overlap between areas identified as critical for the 12 local NCP versus the 2 global NCP, we calculate the area (globally and by country; Supplementary Data 1) where local NCP are selected by the optimization and global NCP are not, where global NCP are selected by the optimization and local NCP are not, and where both are selected by their respective optimizations (the overlap). To calculate the species and languages represented by critical natural assets, we count the number of species whose AOH area targets overlaps these areas, according to both log-linear representation targets (Supplementary Data 3) and targets based on IUCN Red List criteria (Supplementary Data 4), and the number of languages partially intersecting these areas (Supplementary Data 5) globally and within each country (for more detail, see Supplementary Note 1 Section 23).

## Reporting summary

Further information on research design is available in the Nature Portfolio Reporting Summary linked to this article.

## Data availability

All final data outputs are available on Open Science Framework[72] (https://osf.io/r5xz7/).

## Code availability

Code to generate optimization results is also available on Open Science Framework[72] (https://osf.io/r5xz7/).

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

## Acknowledgements

We thank all the participants of two working groups hosted by Conservation International and the Natural Capital Project for their insights and intellectual contributions. For further advice or assistance, we thank A. Adams, K. Brandon, K. Brauman, A. Cramer, G. Daily, J. Fisher, R. Gould, L. Mandle, J. Montgomery, A. Rodewald, D. Rossiter, E. Selig, A. Vogl and T. M. Wright. The two working groups that provided the foundation for this analysis were funded by support from the Marcus and Marianne Wallenberg Foundation to the Natural Capital Project (R.C.-K. and R.P.S.) and the Betty and Gordon Moore to Conservation International (R.A.N. and P.M.C.).

## Author contributions

R.C.-K., R.P.S., M.M., E.F.-C., L.J.G., P.W.K., K.L.-W., P.B.M., M.N., C.R.L., P.R.R., M.S., J.A.J., A.v.S. and R.A.W. provided data, R.S. and M.S.-M. performed the optimizations, R.C.-K., R.P.S., L.J.G. and P.R.R. performed additional analyses, C.B., S.D., U.P., G.S.-T., M.R.S., C.M.K. and W.R.T. provided framing for policy relevance, R.C.-K., R.A.N., P.M.C., S.P. and D.H. directed the project, R.C.-K. and R.A.N. wrote the first draft of the paper and all authors contributed to revisions.

## Competing interests

The authors declare no competing interests.

## Additional information

**Extended data** is available for this paper at https://doi.org/10.1038/s41559-022-01934-5.

**Correspondence and requests for materials** should be addressed to Rebecca Chaplin-Kramer.

[1]SPRING, Oakland, CA, USA. [2]Institute on the Environment, University of Minnesota, St. Paul, MN, USA. [3]Natural Capital Project, Stanford University, Stanford, CA, USA. [4]Dept. of Natural Resources & Environment, Cornell University, Ithaca, NY, USA. [5]Conservation International, Arlington, VA, USA. [6]Dept. of Applied Economics, University of Minnesota, St. Paul, MN, USA. [7]Dept. of Biology, 1125 Colonel By Drive, Carleton University, Ottawa, ON, Canada. [8]Nature Conservancy of Canada, Toronto, Ontario, Canada. [9]Cornell Lab of Ornithology, Ithaca, NY, USA. [10]Dept. of Geography, King's College London, Bush House, London, UK. [11]World Resources Institute, Washington, DC, USA. [12]Consejo Nacional de Investigaciones Científicas y Técnicas, Instituto Multidisciplinario de Biología Vegetal (IMBIV), CONICET, Casilla de Correo 495, Córdoba, Argentina. [13]Universidad Nacional de Córdoba, Facultad de Ciencias Exactas, Físicas y Naturales, Departamento de Diversidad Biológica y Ecología, Córdoba, Argentina. [14]Dept. of Earth System Science, Stanford University, Stanford, CA, USA. [15]Dept. of Landscape Architecture, Penn State University, University Park, PA, USA. [16]Global Protect Oceans, Lands and Waters Program, The Nature Conservancy, Fort Collins, CO, USA. [17]School of Global Environmental Sustainability, Colorado State University, Fort Collins, CO, USA. [18]The Nature Conservancy, 4245 Fairfax Drive, Arlington, VA, USA. [19]Basque Centre for Climate Change, Sede Building 1, 1st floor. Scientific Campus of the University of the Basque Country, Leioa, Spain. [20]Basque Foundation for Science, Ikerbasque, Bilbao, Spain. [21]Centre for Development and Environment, University of Bern, Bern, Switzerland. [22]College of the Environment, Western Washington University – Salish Sea Region, Everett, WA, USA. [23]SYSTEMIQ Ltd, 110 High Holborn, London, UK. [24]World Wildlife Fund, San Francisco, CA, USA. [25]Dept. of Physical, Earth, and Environmental Sciences, University of Siena, Pian dei Mantellini, Siena, Italy. [26]UN Environment World Conservation Monitoring Centre, Cambridge, UK. [27]Institute for Marine and Antarctic Studies, University of Tasmania, 20 Castray Esplanade, Battery Point, Hobart, Tasmania, Australia. ✉e-mail: rchaplin@umn.edu

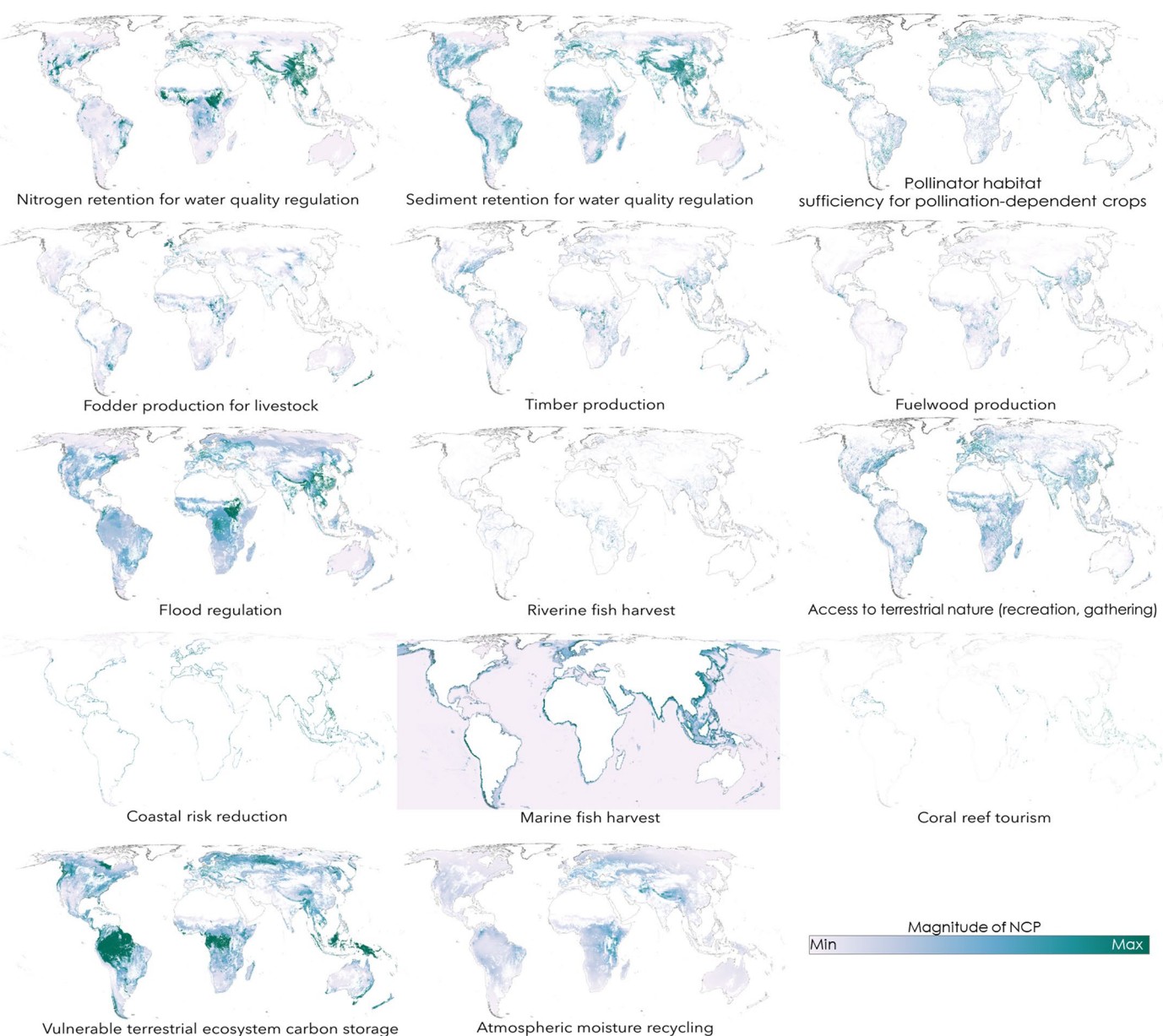

**Extended Data Fig. 1 | Individual maps for the 14 of Nature's Contributions to People (NCP) included in critical natural assets.** Full size maps for each NCP are available at https://osf.io/5nfje.

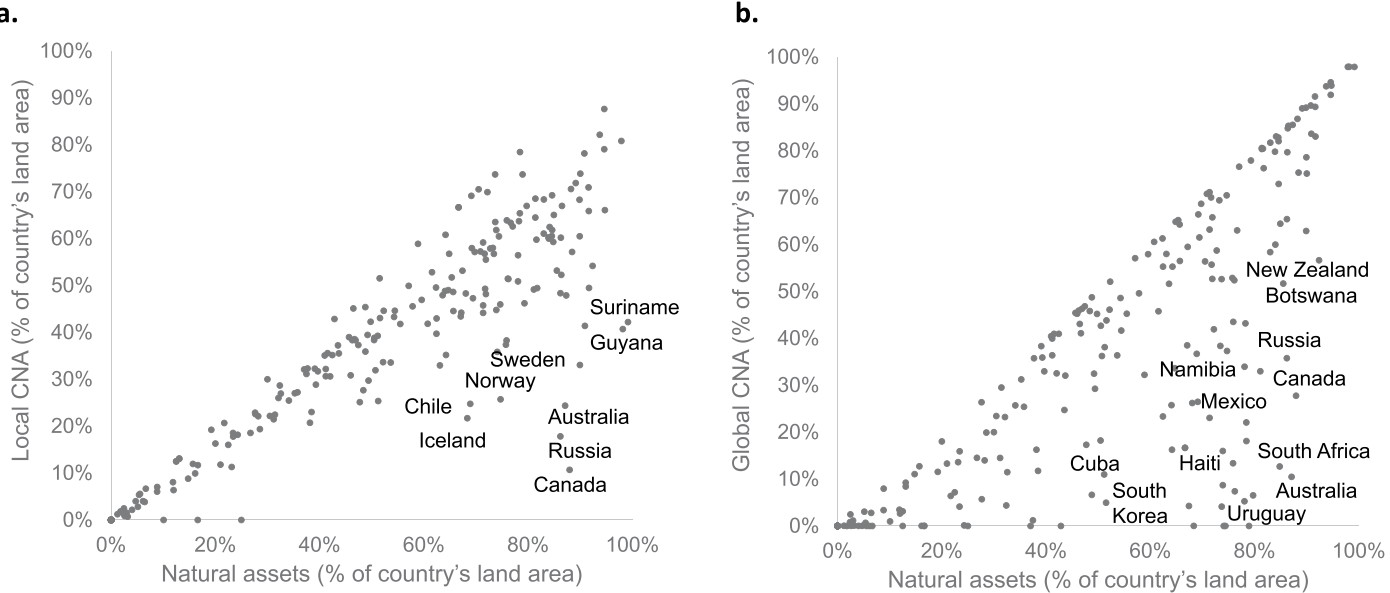

**Extended Data Fig. 2 | Percent of land in critical natural assets (CNA) for local (a) and global (b) benefits, plotted against the percentage of total natural assets in a country.** Labeled countries are outliers, with relatively low critical natural assets area requirements for the percent of total land area made up by natural assets.

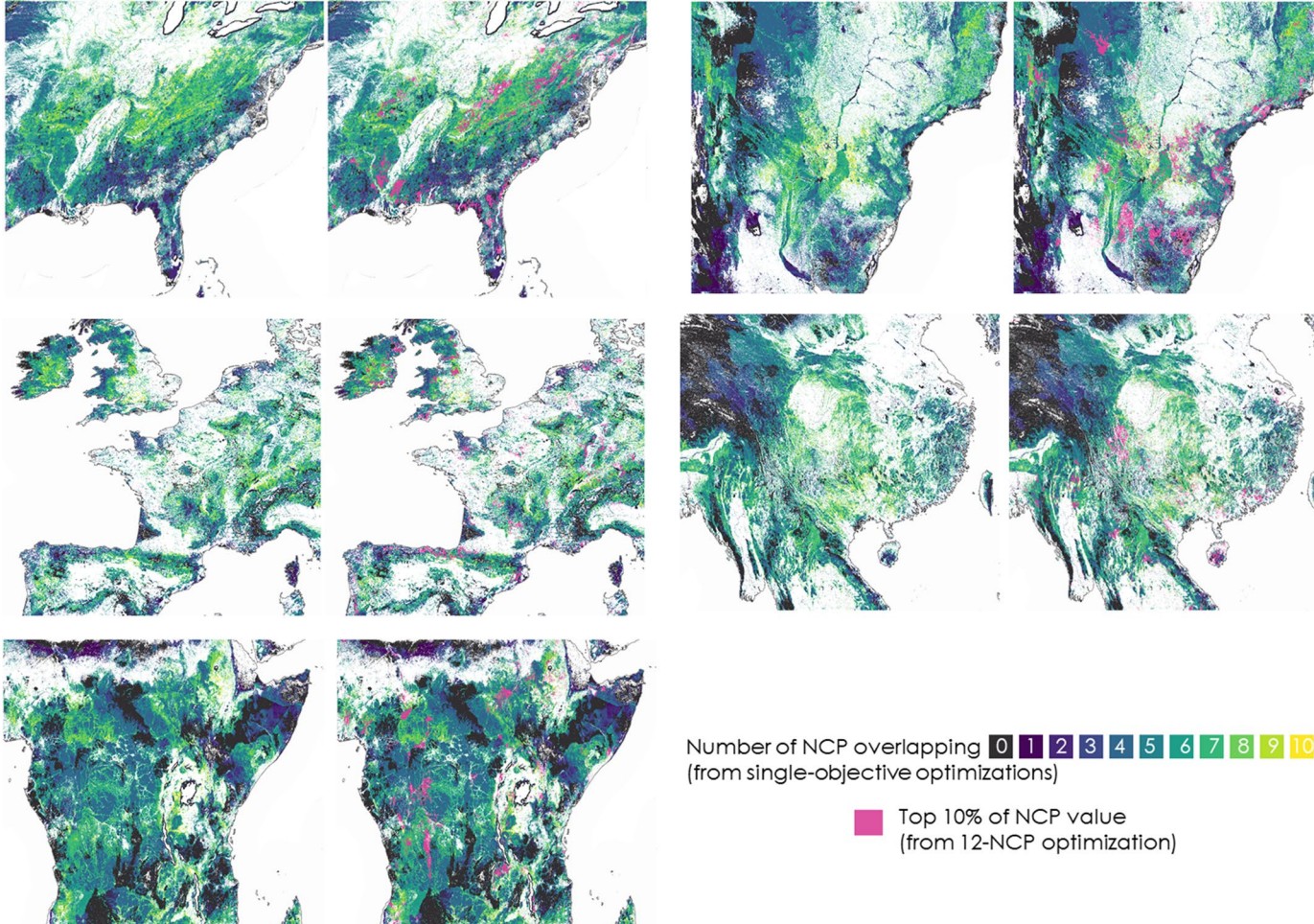

**Extended Data Fig. 3 | Spatial congruence between NCP aligning with critical natural asset hotspots.** Close-ups of regions with high congruence include the southeastern US (top left), central South America (top right), western Europe (middle left), China and southern Asia (middle right) and central Africa (bottom left). Warmer colors (greens and yellow) represent a larger number of overlaps between single-NCP optimizations, while cooler colors (blues and purples) denote fewer overlaps. Pink areas, shown overlaid on the maps on the right of each pair, show the top 10% of highest value areas in the multi-NCP optimization (including all 12 'local' NCP).

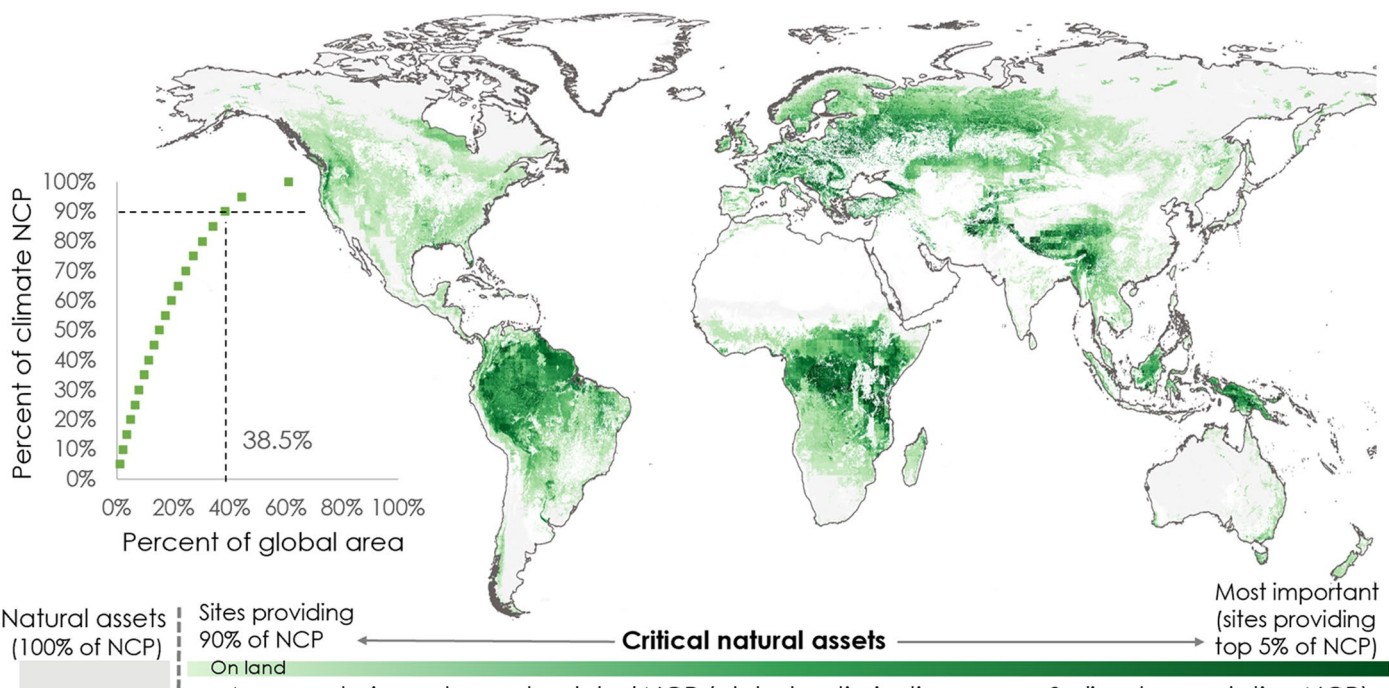

**Extended Data Fig. 4 | Critical natural assets identified through optimization at the global level of two climate-relevant NCP: vulnerable carbon and vegetation-mediated atmospheric moisture regulation.** As in Fig. 1, the NCP accumulation curve reflects the total area required to maintain target levels of both global NCP (optimized globally, not within each country), with dotted lines denoting the area of critical natural assets (90% of global climate NCP in 39% of land area). The map shows critical natural assets for global climate NCP, with darker shades connoting greater contribution to aggregate NCP.

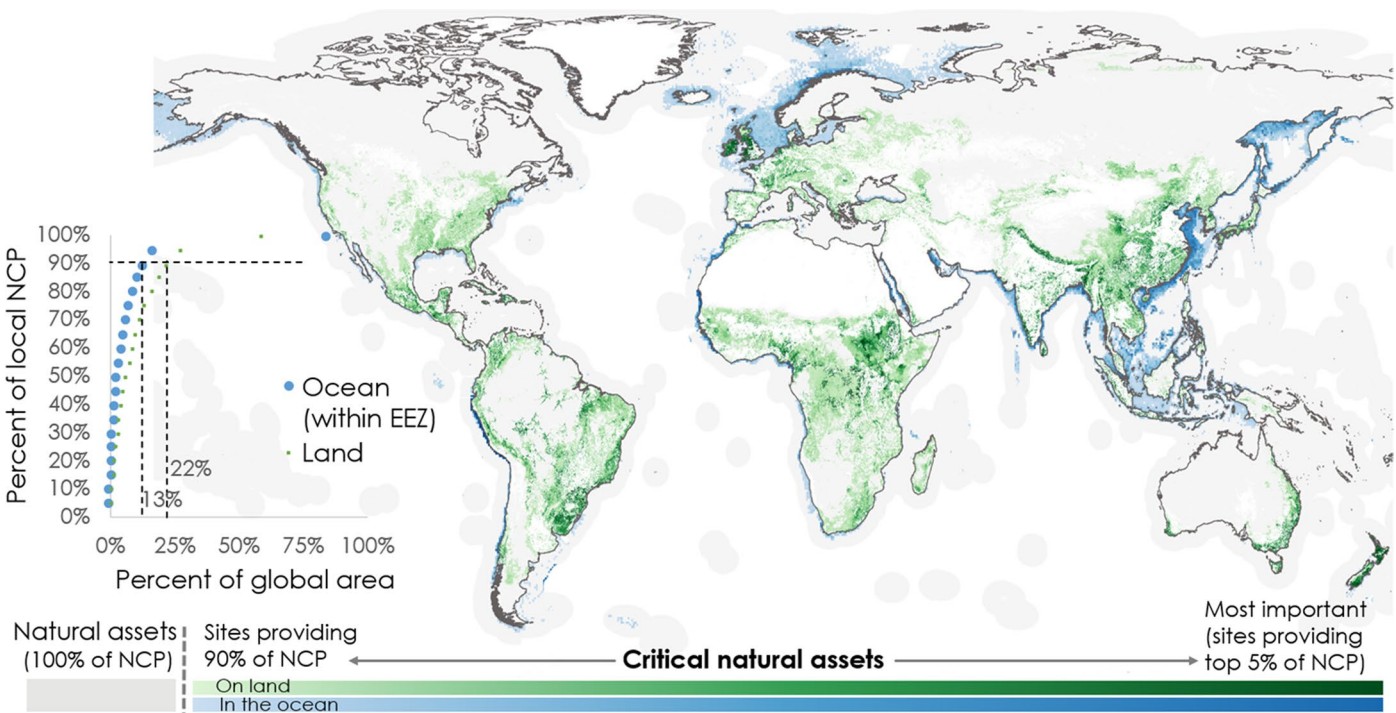

**Extended Data Fig. 5 | Critical natural assets identified through optimization at the global level of 12 'local' NCP.** As in Fig. 1, the NCP accumulation curve reflects the total area required to maintain target levels of all 12 local NCP (but in this case optimized globally, not within each country), with dotted lines denoting the area of critical natural assets (90% of the 12 NCP listed in Fig. 1a in 22% of land area and 13% of EEZ areas). The map shows critical natural assets for local NCP, with darker shades connoting greater contribution to aggregate NCP.

# Reporting Summary

## Statistics

For all statistical analyses, confirm that the following items are present in the figure legend, table legend, main text, or Methods section.

| n/a | Confirmed | |
|---|---|---|
| ☒ | ☐ | The exact sample size (*n*) for each experimental group/condition, given as a discrete number and unit of measurement |
| ☒ | ☐ | A statement on whether measurements were taken from distinct samples or whether the same sample was measured repeatedly |
| ☒ | ☐ | The statistical test(s) used AND whether they are one- or two-sided *Only common tests should be described solely by name; describe more complex techniques in the Methods section.* |
| ☒ | ☐ | A description of all covariates tested |
| ☒ | ☐ | A description of any assumptions or corrections, such as tests of normality and adjustment for multiple comparisons |
| ☒ | ☐ | A full description of the statistical parameters including central tendency (e.g. means) or other basic estimates (e.g. regression coefficient) AND variation (e.g. standard deviation) or associated estimates of uncertainty (e.g. confidence intervals) |
| ☒ | ☐ | For null hypothesis testing, the test statistic (e.g. *F*, *t*, *r*) with confidence intervals, effect sizes, degrees of freedom and *P* value noted *Give P values as exact values whenever suitable.* |
| ☒ | ☐ | For Bayesian analysis, information on the choice of priors and Markov chain Monte Carlo settings |
| ☒ | ☐ | For hierarchical and complex designs, identification of the appropriate level for tests and full reporting of outcomes |
| ☒ | ☐ | Estimates of effect sizes (e.g. Cohen's *d*, Pearson's *r*), indicating how they were calculated |

*Our web collection on statistics for biologists contains articles on many of the points above.*

## Software and code

Policy information about availability of computer code

| Data collection | We used InVEST version 3.8 (free and open-source, available at naturalcapitalproject.stanford.edu/software/invest) and Co$tingNature version 3 (free for non-commercial uses, available at www.policysupport.org/costingnature), along with several open-source pre-published datasets. |
|---|---|
| Data analysis | We used prioritizr available at https://prioritizr.net/ |

For manuscripts utilizing custom algorithms or software that are central to the research but not yet described in published literature, software must be made available to editors and reviewers. We strongly encourage code deposition in a community repository (e.g. GitHub). See the Nature Portfolio guidelines for submitting code & software for further information.

## Data

Policy information about availability of data

All manuscripts must include a data availability statement. This statement should provide the following information, where applicable:
- Accession codes, unique identifiers, or web links for publicly available datasets
- A description of any restrictions on data availability
- For clinical datasets or third party data, please ensure that the statement adheres to our policy

All data and code to run it are available at: https://osf.io/r5xz7/?view_only=d611a688525f4ceb8db4ef4e7528b0e8

# Field-specific reporting

Please select the one below that is the best fit for your research. If you are not sure, read the appropriate sections before making your selection.

☐ Life sciences ☐ Behavioural & social sciences ☒ Ecological, evolutionary & environmental sciences

For a reference copy of the document with all sections, see nature.com/documents/nr-reporting-summary-flat.pdf

# Ecological, evolutionary & environmental sciences study design

All studies must disclose on these points even when the disclosure is negative.

| | |
|---|---|
| Study description | We ran spatially explicit models of nature's contributions to people and integrated their outputs along with pre-existing datasets into a multi-objective optimization. |
| Research sample | Spatially-explicit GIS data |
| Sampling strategy | Data were resampled from 300 m resolution to 2 km resolution to include in the spatial optimization; otherwise no sub-sampling was performed. |
| Data collection | Data were provided by coauthors Chaplin-Kramer, Sharp, Mulligan, McIntyre, Spalding, Watson, Keys, and Roehrdanz. Methods to produce each spatial dataset are detailed in the Supplementary Information |
| Timing and spatial scale | Data are mostly from 2015, but span the 2000-2017 window (annual/snapshot), and are global in extent. |
| Data exclusions | No data were excluded |
| Reproducibility | Code is published to allow reproducibility |
| Randomization | Randomization is not necessary because all data are included in the optimization |
| Blinding | Blinding is not relevant because all data are included in the optimization |

Did the study involve field work? ☐ Yes ☒ No

# Reporting for specific materials, systems and methods

We require information from authors about some types of materials, experimental systems and methods used in many studies. Here, indicate whether each material, system or method listed is relevant to your study. If you are not sure if a list item applies to your research, read the appropriate section before selecting a response.

## Materials & experimental systems

| n/a | Involved in the study |
|---|---|
| ☒ | ☐ Antibodies |
| ☒ | ☐ Eukaryotic cell lines |
| ☒ | ☐ Palaeontology and archaeology |
| ☒ | ☐ Animals and other organisms |
| ☒ | ☐ Human research participants |
| ☒ | ☐ Clinical data |
| ☒ | ☐ Dual use research of concern |

## Methods

| n/a | Involved in the study |
|---|---|
| ☒ | ☐ ChIP-seq |
| ☒ | ☐ Flow cytometry |
| ☒ | ☐ MRI-based neuroimaging |

