## [Peer Review File · Nature Ecology & Evolution]

Peer Review Information

Journal: Nature Ecology & Evolution

Manuscript Title: Mapping the planet's critical natural assets

Corresponding author name(s): Rebecca Chaplin-Kramer

Editorial Notes:

Reviewer Comments & Decisions:

Decision Letter, initial version:
--

9th June 2022

Dear Dr Chaplin-Kramer,

Your manuscript entitled "Mapping the planet's critical natural assets" has now been seen by three reviewers, whose comments are copied below. The reviewers have raised a number of concerns which will need to be addressed in a revised manuscript before we can offer publication in Nature Ecology & Evolution. In particular, while Reviewers 1 and 3 are positive, Reviewer 2 is skeptical of the approach and raises concerns about the validity of some of the ecosystem service estimates (in particular pollination, but the concerns may apply to other services).

We invite you to revise your manuscript taking into account all reviewer comments. Please highlight all changes in the manuscript text file.

* If you have not done so already please begin to revise your manuscript so that it conforms to our Article format instructions at <http://www.nature.com/natecolevol/info/final-submission>. Refer also to any guidelines provided in this letter.

[REDACTED]

Nature Ecology & Evolution is committed to improving transparency in authorship. As part of our efforts in this direction, we are now requesting that all authors identified as 'corresponding author' on published papers create and link their Open Researcher and Contributor Identifier (ORCID) with their account on the Manuscript Tracking System (MTS), prior to acceptance. ORCID helps the scientific community achieve unambiguous attribution of all scholarly contributions. You can create and link your ORCID from the home page of the MTS by clicking on 'Modify my Springer Nature account'. For more information please visit www.springernature.com/orcid.

[REDACTED]

Reviewer expertise:

Reviewer #1: Ecosystem services

Reviewer #2: Mapping, spatial prioritizations

Reviewer #3: Modeling ecosystem services

Reviewers' comments:

Reviewer #1 (Remarks to the Author):

This is a ground-breaking study that maps out the global natural assets hotspots for ecosystem services or Nature's Contributions to People (NCP). The main findings are quite important, with profound policy implications at the global scale. About 30% of the global land area and 24% of national territorial waters maintain 90% of the total amount of 12 types of NCP. Importantly, these hotspots contain both high levels of biodiversity and cultural diversity (particularly diverse indigenous and native communities). Thus, this study provides empirical evidence that development, climate and conservation goals should and can be synergistically achieved globally. The supplemental material is well prepared, providing essential details about the enormous data sources and painstaking statistical/modeling analysis. All that said, I do have some suggestions for further improvements:

1. Does the term "critical natural assets" accurately convey what you really mean? It seems to me that "natural asset hotspots" would be more accurate and self-explanatory because "hotspots" has a sense of spatial location/place. "Critical natural assets" can be easily confused with "Critical Natural Capital". The latter has been widely used in ecological/environmental economics and sustainability research, but has a quite different meaning (see papers by Ekins et al.).
2. Like many others, I'm not a fan of the term Nature's Contributions to People (NCP) because it does not really differ that much from the term ecosystem services (ES) which has long been used globally. I would use ES, but I know that IPBES folks insist on using NCP, no matter what. But it's a bit odd that you're using InVEST on the one hand, and call its outputs NCP, on the other hand.
3. What about the non-critical natural assets? Although grasslands are mentioned, it's worrisome to see that most global drylands are mapped as "non-critical natural assets". But drylands, including vast deserts around the world, are unique and valuable ecosystems, too, although they are not biologically as diverse as forests. They cover 41% of the global land area and are home to about 35% global population. Many indigenous and native peoples and poor nations reside in arid regions. Global sustainability cannot be achieved without taking care of drylands. I suggest that the authors make some explicit comments on this issue.
4. The main text lacks clear organization or structure. Adding a few headings would be helpful. Also, the term critical natural assets is defined multiple times throughout the paper, which is not necessary.
5. Not everyone will read the supplemental material, so some important information, such as the spatial resolution of data/pixel size and key findings of scale sensitivity, should be included in the main text.
6. Related to point 5 above, it should be stated clearly in the main text if the critical natural assets include nature reserves and protected areas.

37. Clarification on 90% of the total 12 local NCP – does this mean each of the 12 NCP is to be maintained at/above the level of 90%? If it's for the total amount of all 12 NCP lumped together, is it possible that some of them may be maintained at a level that is much lower than 90%?

Reviewer #2 (Remarks to the Author):

Major comments

This study aims to identify the areas that, if conserved, would maintain critical natural assets, defined as 90% of the total global provision of 12 different Nature Contributions to People. This study also seeks to investigate the extent to which NCP provision can be jointly optimized, the level of co-benefits for selected terrestrial vertebrates of NCP-optimized conservation areas and the number of people living in different sets of priorities for conserving NCP provision.

From a policy perspective I understand the need for such analysis, especially in the light of the upcoming negotiations for the Global Biodiversity Framework. However I do have some perplexities about the methods and related assumptions.

I am not an expert of ecosystem service modelling and therefore I cannot comment on the quality of the models underlying much of the analyses done here, and I strongly recommend identifying few reviewers able to judge the appropriateness of the NCP models, e.g. for pollination, timber production, fuel production, flood regulation and other local NCPs. I suspect that it does not make sense to map these NCP globally using simple proxies, mostly based on global land-cover maps.

I can say with absolute confidence that calculating the amount of "natural habitat" in a radius of 2km around a cropland pixel is in no way sufficient to model pollination service. Pollination mostly depends on several other factors within and outside cultivated areas, these includes the type, timing and quantity of pesticides applications, the presence of natural landscape features and the spatial configuration of them within managed landscapes. None of them are accounted for in pollination models. The type and intensity of management of grassland, cropland and forest to affect other NCPs modelled here, and these too are largely ignored by these models.

In summary, the pollination service model, like many other NCP models from InVEST, WaterWorld or Co\$ting Nature, are reclassifications of the ESA CCI land-cover map with the addition of some other spatial data. This makes them hardly credible both in terms of spatial location of priorities (Fig 1c and Fig 3), the amount of people benefiting (Fig 2), and the extent of NCP provision that can be obtained by any given set of areas (Fig 1b).

I am not sure that attempting to provide measures of the extent of NCP provision retained through habitat conservation from global analyses and with such model is a valid endeavor.

Representation targets for species. You used the log-linear interpolation of species ranges that many before you have used, but that has no ecological foundation behind it. It was coined in 2004 and by own admission of the authors it was chosen because it made plausible sense, but there was really no empirical evidence behind it. I would like to see some justification for these approaches (besides

4stating that others before having used them), or the use alternative methods for setting species representation targets that are based on empirical evidence of their sufficiency, e.g. using Red List Criteria.

Targets for ecosystems. I understand that it is quite difficult to estimate what is a sufficient amount of ecosystem service provision, and therefore you set incremental targets as a proportion of the total potential provision of NCP at the national or global level. This, however, affects your conclusion about the 30% or 44% area requirement, because you choose to focus on the 90% provision. You justify this because of the diminishing returns beyond this target level, but what if 70% or some other target level was enough? I suggest adding some text on the implications of your choice of focusing on 90% for the critical assets definition, ideally with some thoughts on how these could be overcome, i.e. what information would you need to define a target level of NCP?

Other comments

In various parts of the supplementary materials the fonts type and size changes, as if the text was copy-pasted and not checked.

The text at lines 1022-1040 would be best placed under data limitations

Lines 130-135 These statistics on overlap are not clear. E.g. endemic, are you referring to country endemics? Or other administrative or biogeographic unit? Also, for the overlap, does any extent of overlap counts towards these statistics?

Line 930. This is not the definition of a minimum set problem. The correct definition (in a spatial planning context) is the set of grid cells that (if protected) satisfies all conservation targets at a minimum cost (in this case area). You formally describe this mathematically at line 932, but the text above it describes two maximizations and no constraints, which is A) impossible in linear programming B) not the minimum set problem C) not what the equation describes

The extended data figures seem to be repeated, first in small size then in full page. In addition, the legend is missing and at least 1 figure has multiple colors which would require explanation (access to nature).

Reviewer #3 (Remarks to the Author):

While there are a number of 'prioritisation' approaches published in the literature, the work presented by Chapin-Kramer and colleagues is, to the best of my knowledge, the most comprehensive global approach at mapping important areas for a number of NCP. The work is well written, easy to follow and the methods are clearly documented in the main text and in the SI material. Furthermore, I think the authors have appropriately tested the sensitivity of their optimisation approach, as documented in section 22 of the SI material. I would recommend this manuscript for publication and have only a couple of minor comments to address.

51. While reviewing the manuscript I felt there was a lack of discussion about how nature can be valued for reasons beyond it's 'usefulness' to humans (a common criticism of NCP). The authors actually do discuss this in the SI material, starting L1142. I think it would be good to move this particular section, or at least some of it, to the main text.

2. Reference 25, O'Connor et al 2021, use an overlap approach at a regional scale for Europe, however in that study the authors find little overlap between priority areas for biodiversity and cultural and regulating NCP. This is somewhat at odds with the findings in this manuscript. Can the authors add an explanation as to why the results of the two studies differ so? Could it be argued that by aggregating the country level results at a global level (as done here) we risk overlooking important regional differences/priorities?

3. There is a typo in panel b of figure 1 'percnt' should be 'percent'.

*****END*****

Author Rebuttal to Initial comments

Reviewer #1 (Remarks to the Author):

This is a ground-breaking study that maps out the global natural assets hotspots for ecosystem services or Nature's Contributions to People (NCP). The main findings are quite important, with profound policy implications at the global scale. About 30% of the global land area and 24% of national territorial waters maintain 90% of the total amount of 12 types of NCP. Importantly, these hotspots contain both high levels of biodiversity and cultural diversity (particularly diverse indigenous and native communities). Thus, this study provides empirical evidence that development, climate and conservation goals should and can be synergistically achieved globally. The supplemental material is well prepared, providing essential details about the enormous data sources and painstaking statistical/modeling analysis. All that said, I do have some suggestions for further improvements:

We thank the reviewer for these kind words and constructive suggestions.

61. Does the term “critical natural assets” accurately convey what you really mean? It seems to me that “natural asset hotspots” would be more accurate and self-explanatory because “hotspots” has a sense of spatial location/place. “Critical natural assets” can be easily confused with “Critical Natural Capital”. The latter has been widely used in ecological/environmental economics and sustainability research, but has a quite different meaning (see papers by Ekins et al.)

*We understand your concern about using the word “critical” and we thought about many other words to describe our main concept in the paper. “Hotspots” have some appeal because, as you point out, it conveys a sense of location/place. In fact, in the previous draft we had used the term “hotspots” to refer to the places that delivered the top 5 to 10% of NCP in the least land area. To avoid confusion, however, in the revision we have now changed “hotspots” to “high value areas”, which also conveys a sense of location/place and clearly says that these are the places that contribute the most to NCP per unit area. We prefer the term “critical natural assets” to “hotspots” for the specific 90% threshold we apply, as it better reflects the motivation for this work to identify natural ecosystems that cannot be easily substituted for or replaced. In fact, we originally set out to identify “critical natural capital” just as Ekins defined it -- identifying areas where ecosystems provide important environmental functions that cannot be substituted in the provision of these functions by manufactured capital. We also appreciate Brand’s (2009) synthesis of multiple possible dimensions of “criticality” as it applies to natural capital (socio-cultural, ecological, sustainability, ethical, economic, and human survival; *Ecological Economics* 68: 605–612). We conducted a literature search and could not find many examples of how this conceptual definition of critical natural capital had actually been operationalized or applied (a few examples of studies that claim to have done so (e.g., MacDonald et al. 1999 in Scotland, Lu et al. 2017 in China, Chen et al. 2020, Ferrari et al. 2012 in France) used proxy, non-spatial indicators related to environmental quality, not ecosystem services). To our knowledge, no one has attempted this at a global scale. However, after conversations with several economists, we realized that getting at non-substitutability would be very challenging. We thus changed the term from “capital” to “assets” to avoid confusion with the formal economic usage of “critical natural capital”. Once we had developed data on the relationship between land (or ocean) area and nature’s contributions to people, we discovered that NCP-area curves had a natural inflection point where adding additional units of land (or ocean) provided sharply diminishing returns. We felt that this inflection point could be considered a useful indicator of “critical” natural assets, as the lands and waters up until this point provide NCP that could not be duplicated elsewhere without requiring significantly more area, and in some cases could not be duplicated. Beyond the inflection point, additional area provides relatively little additional NCP and therefore could be considered less critical. So we adopted this as our new, albeit imperfect, definition of criticality. We added language explaining our choice of terms in the introduction (L 62–67) and acknowledging in the discussion that it is a first attempt to begin to operationalize the concept of criticality at a global scale, but that future work is needed to refine our estimates (L246–249).*

72. Like many others, I'm not a fan of the term Nature's Contributions to People (NCP) because it does not really differ that much from the term ecosystem services (ES) which has long been used globally. I would use ES, but I know that IPBES folks insist on using NCP, no matter what. But it's a bit odd that you're using InVEST on the one hand, and call its outputs NCP, on the other hand.

Again, we definitely understand this concern. In the IPBES framework, ecosystem services are one dimension of Nature's Contributions to People, and our use of the latter term is meant to be more inclusive to the audiences for whom the term "ecosystem services" doesn't resonate and to link better to the Convention on Biodiversity, which also uses the term NCP. But we appreciate that many readers are still confused by the distinction and to avoid this we have now added text in L 48-51 to specifically acknowledge that these are ecosystem services, and that ecosystem services are in fact NCP (even if the broad definition of NCP is also more inclusive of e.g., relational values than traditional definitions of ecosystem services), but that we use the term NCP to better align with international frameworks and policy.

3. What about the non-critical natural assets? Although grasslands are mentioned, it's worrisome to see that most global drylands are mapped as "non-critical natural assets". But drylands, including vast deserts around the world, are unique and valuable ecosystems, too, although they are not biologically as diverse as forests. They cover 41% of the global land area and are home to about 35% global population. Many indigenous and native peoples and poor nations reside in arid regions. Global sustainability cannot be achieved without taking care of drylands. I suggest that the authors make some explicit comments on this issue.

This is an excellent point and we agree with the reviewer that global drylands are important to many people and provide many NCP. Unfortunately, global data on the NCP provided by drylands are extremely limited; we do not have any global maps of specific NCP that are unique to drylands such as wind erosion regulation. We do have data for NCP like grazing that occur in many drylands classified in ESA as "sparse", "shrub" or "grassland". However, we are not able to include in our analysis areas classified in ESA as "bare" (like the Sahara, much of the Middle East, the Gobi desert). These are shown in Fig 1 in white, which denotes areas not included in the analysis (not in grey, which are the natural assets the reviewer points out as not deemed critical). We previously acknowledged the data limitations that prevented us from including cropland and urban areas in our analysis of natural assets; we now extend this caveat to deserts (as well as arctic systems, which arguably suffer from the same data limitations and underrepresentation in the literature) in L 211-213.

4. The main text lacks clear organization or structure. Adding a few headings would be helpful. Also, the term critical natural assets is defined multiple times throughout the paper, which is not necessary.

We appreciate this helpful feedback; we have eliminated redundancy in our definition of critical natural assets and added the following headings: Extent and location of critical natural assets, Number of people benefitting from critical natural assets, Overlaps between local and global priorities, Gaps and next steps, and Conclusions.

5. Not everyone will read the supplemental material, so some important information, such as the spatial resolution of data/pixel size and key findings of scale sensitivity, should be included in the main text.

We now include the spatial resolution of each dataset in its respective section of the methods, as well as the resolution at which the optimization was conducted within that section of the Methods. We state the resolution of the optimization in the main text (leaving aside the individual datasets, since their varying resolutions would add too much complexity there) and also include a description of the findings on scale sensitivity in the section commenting on uncertainty.

6. Related to point 5 above, it should be stated clearly in the main text if the critical natural assets include nature reserves and protected areas.

Thank you for this suggestion. Nature reserves and protected areas are included in our areas eligible for prioritization and are not differentiated or treated differently from lands that are not nature reserves or protected areas. We calculated and now include mention of how much of critical natural assets are located within protected areas (according to overlap with the World Database on Protected Areas) in L 125-126.

7. Clarification on 90% of the total 12 local NCP – does this mean each of the 12 NCP is to be maintained at/above the level of 90%? If it's for the total amount of all 12 NCP lumped together, is it possible that some of them may be maintained at a level that is much lower than 90%?

Thank you for pointing out that this needed clarification. The optimization indeed ensures that the 90% target is met for each of the 12 NCP, meaning it's possible that some slightly exceed, but none fall short of that target, and we now clarify this point on L 60.

Reviewer #2 (Remarks to the Author):

Major comments

This study aims to identify the areas that, if conserved, would maintain critical natural assets, defined as 90% of the total global provision of 12 different Nature Contributions to People. This study also seeks to investigate the extent to which NCP provision can be jointly optimized, the level of co-benefits for selected terrestrial vertebrates of NCP-optimized conservation areas and the number of people living in different sets of priorities for conserving NCP provision.

From a policy perspective I understand the need for such analysis, especially in the light of the upcoming negotiations for the Global Biodiversity Framework. However I do have some perplexities about the methods and related assumptions.

We appreciate this feedback to improve the manuscript and clarify its methods and findings.

I am not an expert of ecosystem service modelling and therefore I cannot comment on the quality of the models underlying much of the analyses done here, and I strongly recommend identifying few reviewers able to judge the appropriateness of the NCP models, e.g. for pollination, timber production, fuel production, flood regulation and other local NCPs. I suspect that it does not make sense to map these NCP globally using simple proxies, mostly based on global land-cover maps.

We fully acknowledge the limitations of trying to map these NCP globally, but these models all include far more information than simple land-cover proxies. Timber and fuelwood production, for example, use remotely-sensed dry matter productivity data, which varies within land-cover types. Nitrogen and sediment retention is modeled with InVEST and flood regulation is modeled with WaterWorld, which are all fairly complex hydrologic models that use climatic, soil, topography, and other datasets in addition to land cover, in a series of process-based functions that are routed through the landscape. Furthermore, many of the models included here have already been peer-reviewed and published elsewhere; they simply have never before been brought together in the same place:

- nitrogen, pollination, coastal protection (Chaplin-Kramer et al. 2019)

10- riverine fish catch (McIntyre et al. 2016)
- marine fish catch (Watson and Tidd 2018)
- coral reef tourism (Spalding et al. 2017)
- flood regulation (Gunnell et al. 2019)
- atmospheric moisture recycling (Keys et al. 2016)
- carbon (Noon et al. 2021)

For the new models that we introduce in this paper and that have not been published at a global scale before now (timber, fuelwood, sediment retention, access to nature), we have attempted to clearly explain the modeling assumptions, input datasets, and limitations in the supplemental materials. While we appreciate that all of these models could be improved with more accurate data and more realistic representation of local context if run at a local or regional scale, there is great demand for global estimates and we believe these models provide the best available information at that scale. We originally tried to provide a brief summary of each model in the methods included in the main text with reference to where they are more fully documented in the SI. However, to more clearly convey that these models are not simply land-cover proxies, we have now brought a bit more of this detail into the main methods by outlining the data sources used by each model.

I can say with absolute confidence that calculating the amount of “natural habitat” in a radius of 2km around a cropland pixel is in no way sufficient to model pollination service. Pollination mostly depends on several other factors within and outside cultivated areas, these includes the type, timing and quantity of pesticides applications, the presence of natural landscape features and the spatial configuration of them within managed landscapes. None of them are accounted for in pollination models. The type and intensity of management of grassland, cropland and forest to affect other NCPs modelled here, and these too are largely ignored by these models.

We fully agree that our attempt to capture areas of importance for crop pollination falls short of representing all the factors affecting pollinators. However, as we are presenting a global analysis, we are limited to globally available datasets, and none exist for pesticides or management intensity in non-crop working lands. Our method does go beyond merely calculating the amount of natural habitat. We linearly interpolate a "pollination sufficiency" value with the assumption that if there's 30% of natural habitat within 2km, crops are fully pollinated (based on work by Claire Kremen and others). According to this, 0% natural habitat is 0% sufficient, 30% habitat is 100% sufficient, and 15% habitat is 50%

sufficient, etc. We could have used different shapes of that relationship but we thought anything other than linear needed justification. That sufficiency index is then multiplied by the proportion of yields that are pollination dependent, based on the crop mixes grown there (or the best/most recent estimate of crop mixes for >150 crops, provided by Monfreda et al. 2008). While this treats all natural habitat equally, we don't know of a way that nesting and floral availability could defensibly be differentiated by different natural LULC types globally. Nobody has done that, and we believe that the uncertainties associated with any differences between LULC types could be greater than just treating all habitat equally. However, we do acknowledge that there are many factors missing that prevent a calculation of the full pollination value, as the reviewer points out, and as such we have adjusted the name of this NCP to reflect that it only accounts for the potential of habitat to provide pollinators: we now use the term pollinator habitat sufficiency for pollination-dependent crop production.

In summary, the pollination service model, like many other NCP models from InVEST, WaterWorld or Co\$ting Nature, are reclassifications of the ESA CCI land-cover map with the addition of some other spatial data. This makes them hardly credible both in terms of spatial location of priorities (Fig 1c and Fig 3), the amount of people benefiting (Fig 2), and the extent of NCP provision that can be obtained by any given set of areas (Fig 1b).

I am not sure that attempting to provide measures of the extent of NCP provision retained through habitat conservation from global analyses and with such model is a valid endeavor.

We respectfully disagree. As explained above, almost none of these models are reclassifications of ESA land cover with the addition of some other spatial data. The water-related NCP are derived from process-based hydrologic models. The biomass-related NCP are based on remotely-sensed productivity data. The fisheries NCP come from spatially disaggregated statistical models. The only NCP for which 'reclassification' of ESA is even a partially accurate description is the "nature access" layer, which is based on subsetting out all "natural" and "semi-natural" land cover classes (as described in the text) and a new application of the Weiss et al. friction layer dataset to calculate travel time across pixels to subset out the natural habitat within an hour (or six hour) travel. But even that involved a least-cost travel time algorithm which provides new insights into the accessibility of nature by different populations. Most of the models have been validated in many regions (though not globally) with ground based observations and have been shown to perform reliably at differentiating higher value areas from areas providing lower levels of benefits. We now cite the studies validating the individual models in each model's section of the SI to provide the evidence that these approaches have been well tested and can be interpreted in the ways that we've used here, and highlight sources of uncertainty for each so that the reader can better understand the model assumptions.

Representation targets for species. You used the log-linear interpolation of species ranges that many before you have used, but that has no ecological foundation behind it. It was coined in 2004 and by own admission of the authors it was chosen because it made plausible sense, but there was really no empirical evidence behind it. I would like to see some justification for these approaches (besides stating that others before having used them), or the use alternative methods for setting species representation targets that are based on empirical evidence of their sufficiency, e.g. using Red List Criteria.

We thank the reviewer for raising this valuable point. To clarify, the targets were not used to drive the spatial optimization for NCP, but rather as a post-hoc analysis of the species-level co-benefits of optimal areas for NCP. An analysis of optimal areas for both NCP and biodiversity with species level targets is a topic for future research. The use of the log-linear representation targets was to provide contextual interpretation of the individual species statistics for range intersection with NCP priority areas. We chose the representation target framework indeed because it has often been used in the literature and therefore provides a familiar system to represent the interplay between percentage of species range overlap and overall species range size. For example, the log-linear framework is used in the species protection index which is a proposed indicator under the post-2020 global biodiversity framework (e.g. Jetz, W. et al. 2022, Include biodiversity representation indicators in area-based conservation targets. Nature Ecology & Evolution 6, 123–126). We agree with the reviewer that the log-linear species target framework is imperfect and is an ongoing area of research. Certainly any species-focused action would require a more precise accounting of range sufficiency. We have now supplied the full species level intersection statistics as SI Table 4 so that other targets of range sufficiency may be explored, and report on L 153-155 and 164-165 the results for Red List Criteria as an example of how critical natural assets could be considered alongside different biodiversity targets.

Targets for ecosystems. I understand that it is quite difficult to estimate what is a sufficient amount of ecosystem service provision, and therefore you set incremental targets as a proportion of the total potential provision of NCP at the national or global level. This, however, affects your conclusion about the 30% or 44% area requirement, because you choose to focus on the 90% provision. You justify this because of the diminishing returns beyond this target level, but what if 70% or some other target level was enough? I suggest adding some text on the implications of your choice of focusing on 90% for the critical assets definition, ideally with some thoughts on how these could be overcome, i.e. what information would you need to define a target level of NCP?

It is true that 90% provision is somewhat arbitrary and one could choose 70% or some other number. This is one of the main reasons why we presented Figure 1, which shows the amount of area needed to meet various targets, ranging from 0 to 100% in 5% increments. We chose 90% to serve as a pragmatic approach to maintaining the provision of ecosystem services at close to current levels, recognizing that the final 10% becomes far more costly because of diminishing returns (the accumulation curve in Figure 1 bends significantly at 90%).

We recognize that current levels may not be considered enough in the future, or even right now. There are a variety of issues with defining what constitutes “enough”, which are important grounds for further research, which we now include in the discussion section (L 231-249):

- We don't know how much of the current need for nature is already unmet. In many parts of the world, natural ecosystems are already degraded or converted, so 90% of what is left might actually be much too little, especially for the world's most vulnerable people living in highly degraded areas (places prone to catastrophic flooding due to habitat conversion, or places where farmland or pasture has become too arid or too saline, places where fish populations have crashed).*
- We are unable to predict future demand for NCP due to climate change, population growth, and change in consumption*
- We are not currently able to account for tipping points or planetary boundaries (e.g. how much of a tropical forest can disappear before the whole ecosystem tips into a different state?)*

Therefore we now frame our results as a first attempt to define a minimum set of critical natural assets (L 246-249). We do not believe there is value in exploring lower targets, as it will confuse the message and if anything we may need far more than 90% -- we may need more than 100% of what we have now!

As noted in response to Reviewer 1 above, in trying to address this question of how much natural capital is “enough”, we originally set out to identify “critical natural capital” just as Ekins (2003) defined it – that critical natural capital enables some function to be performed for which there is no substitute that could provide the same function. In our case, we originally sought to identify areas where ecosystems provide important environmental functions that cannot be substituted for by other forms of capital. We conducted a literature search and could not find many examples of how this conceptual definition of critical natural capital had actually been operationalized or applied (a few examples of studies that claim

to have done so (e.g., MacDonald et al. 1999 in Scotland, Lu et al. 2017 in China, Chen et al. 2020, Ferrari et al. 2012 in France) used proxy, non-spatial indicators related to environmental quality, not ecosystem services). To our knowledge, no one has attempted this at a global scale. After conversations with several economists, we realized that getting at non-substitutability would be very challenging and so we dropped this notion of defining critical natural capital. In response to Reviewer 1 above, we have added some material in the introduction about “critical natural capital” in terms of meeting people’s needs and accounting for substitutability from other forms of capital. To address the concerns of Reviewer 2, we have also added text to the discussion reflecting on why accounting for substitutes (overestimating the criticality) may be less of a concern than underestimating, especially since substitutes only tend to substitute for a single NCP and the vast majority of selected pixels provide many (if not most) of the NCP (L235-237).

Other comments

In various parts of the supplementary materials the fonts type and size changes, as if the text was copy-pasted and not checked.

We thank the reviewer for catching this; we have carefully reviewed the entire SI to remove such discrepancies.

The text at lines 1022-1040 would be best placed under data limitations

Thank you for this suggestion; we have moved the text

Lines 130-135 These statistics on overlap are not clear. E.g. endemic, are you referring to country endemics? Or other administrative or biogeographic unit? Also, for the overlap, does any extent of overlap counts towards these statistics?

Because the prioritizations were done by country, we examined the number of species whose ranges fell completely within the country, vs. were shared by other countries. Thus, we were referring to country endemics, and we have now clarified this in L 152. Also, yes, the species range statistics are the total intersection of each species AOH with the NCP priority areas, and we now clarify that in L 148.

Line 930. This is not the definition of a minimum set problem. The correct definition (in a spatial planning context) is the set of grid cells that (if protected) satisfies all conservation targets at a minimum cost (in this case area). You formally describe this mathematically at line 932, but the text above it describes two maximizations and no constraints, which is A) impossible in linear programming B) not the minimum set problem C) not what the equation describes

Thank you for this comment. We completely agree with you that we did not correctly describe what we did in the optimization and we are grateful that you pointed this out. We have rewritten this section to correctly represent the minimum set problem as follows:

The minimum set problem seeks to find the set of planning units that minimizes the overall cost of meeting a set of targets. This problem can be expressed mathematically for a set of planning units (I , indexed by i) and a set of features (J , indexed by j) as:

$$\begin{aligned} & \text{Minimize } \sum_{i=1}^I x_i c_i \\ & \text{subject to} \\ & \sum_{i=1}^I x_i r_{ij} \geq T_j \quad \forall j \in J \end{aligned}$$

Here, x_i is the decision variable (i.e., specifying whether planning unit i has been selected (1) or not (0)), c_i is the cost of selecting planning unit i , r_{ij} is the amount of feature j in planning unit i , and T_j is the target for feature j . In our application, we use the amount of area as the cost.

We solved the minimum set problem in the prioritizr R package that one of the authors is a co-developer of (RS).

The extended data figures seem to be repeated, first in small size then in full page. In addition, the legend is missing and at least 1 figure has multiple colors which would require explanation (access to nature).

We thought some people might like to see all the maps on one page (or 1.5 pages!), while others might want to see them larger so they could better resolve their regions of interest. However, we see that this could be confusing, so we instead provide the jpegs for people to zoom in on the OSF site where the data are hosted, and list this in the caption for the figure. We included the legend at the end of the small-

16format figure, but we agree that it would be useful to have on every individual so we have added it to the individual jpegs. The additional colors in the access to nature map were there in error and have been removed. Thank you very much for your careful attention!

Reviewer #3 (Remarks to the Author):

While there are a number of 'prioritisation' approaches published in the literature, the work presented by Chapin-Kramer and colleagues is, to the best of my knowledge, the most comprehensive global approach at mapping important areas for a number of NCP. The work is well written, easy to follow and the methods are clearly documented in the main text and in the SI material. Furthermore, I think the authors have appropriately tested the sensitivity of their optimisation approach, as documented in section 22 of the SI material. I would recommend this manuscript for publication and have only a couple of minor comments to address.

We thank the reviewer for this encouraging assessment.

1. While reviewing the manuscript I felt there was a lack of discussion about how nature can be valued for reasons beyond its 'usefulness' to humans (a common criticism of NCP). The authors actually do discuss this in the SI material, starting L1142. I think it would be good to move this particular section, or at least some of it, to the main text.

Intrinsic values of nature are out of the scope of this paper, which is explicitly focused on a subset of NCP, which are all anthropocentric by definition. We have noted this now, and have also moved some of this material out of the supplement to address this issue explicitly in L250-258.

2. Reference 25, O'Connor et al 2021, use an overlap approach at a regional scale for Europe, however in that study the authors find little overlap between priority areas for biodiversity and cultural and regulating NCP. This is somewhat at odds with the findings in this manuscript. Can the authors add an explanation as to why the results of the two studies differ so? Could it be argued that by aggregating the country level results at a global level (as done here) we risk overlooking important regional differences/priorities?

17Thanks also for this interesting suggestion. We were also puzzled by the difference in findings, but it is difficult to compare directly because many of the NCP they assessed were so different. We both assessed: carbon sequestration, flood control, and pollination. They also assessed foraging areas for wild foods and nature tourism, which may have some overlaps with fuelwood provisioning or with our “nature access” layer that was meant to serve a catch-all for different uses of nature we weren’t able to capture through the other models. But they also assessed air quality regulation, heritage agriculture, heritage forests, which likely have very different distributions than the rest of the terrestrial NCP we modeled (sediment retention, nitrogen retention, fodder for livestock, riverine fisheries, atmospheric moisture regulation, coastal risk reduction). It’s also possible that by starting with “priority” areas for biodiversity they are locking in areas that are not as important at delivering benefits to people; our analysis doesn’t compare how different priority regions are if you start from the objective of biodiversity vs. the objective of NCP (though we agree this would be interesting grounds for further research-- and we do have another paper in prep to explore this very issue). Rather, we start from the objective of NCP only, and then evaluate how much of vertebrate biodiversity is represented in those areas. We acknowledge there are quite a lot of species not represented in these critical natural assets (>40%!) and those may well be found in areas that are priorities for biodiversity but not NCP. We think this is not so much a global vs. regional distinction (since our prioritization was done at the country scale, ensuring 90% targets were met within each country) as difference in underlying data and objectives for optimization. But we appreciate that our previous mention of the O’Connor study was cursory and it deserved more discussion; we have now added this in L 165-169.

3. There is a typo in panel b of figure 1 ‘percnt’ should be ‘percent’.

Thank you so much for catching this! It has been corrected.

Decision Letter, first revision:

14th September 2022

Dear Dr. Chaplin-Kramer,

Thank you for submitting your revised manuscript "Mapping the planet’s critical natural assets" (NATECOLEVOL-220416312A). It has now been seen again by Reviewers 1 and 3 and their comments are below. The reviewers find that the paper has improved in revision, and therefore we'll be happy in principle to publish it in Nature Ecology & Evolution, pending minor revisions to satisfy the reviewers'

18final requests and to comply with our editorial and formatting guidelines.

[REDACTED]

Reviewer #1 (Remarks to the Author):

The authors have adequately addressed all my comments. The manuscript has been substantially improved in terms of organization, clarity, and scientific rigor. I have no doubt that this work is important and timely, and will be influential in years to come.

Jianguo (Jingle) Wu

Reviewer #3 (Remarks to the Author):

I have no further comments to make on this manuscript. I think the authors have sufficiently addressed my previous comments. I recommended this paper for publication.

Our ref: NATECOLEVOL-220416312A

22nd September 2022

Dear Dr. Chaplin-Kramer,

Thank you for your patience as we've prepared the guidelines for final submission of your Nature

19Ecology & Evolution manuscript, "Mapping the planet's critical natural assets" (NATECOLEVOL-220416312A). Please carefully follow the step-by-step instructions provided in the attached file, and add a response in each row of the table to indicate the changes that you have made. Please also check and comment on any additional marked-up edits we have proposed within the text. Ensuring that each point is addressed will help to ensure that your revised manuscript can be swiftly handed over to our production team.

****We would like to start working on your revised paper, with all of the requested files and forms, as soon as possible (preferably within two weeks). Please get in contact with us immediately if you anticipate it taking more than two weeks to submit these revised files.****

In recognition of the time and expertise our reviewers provide to Nature Ecology & Evolution's editorial process, we would like to formally acknowledge their contribution to the external peer review of your manuscript entitled "Mapping the planet's critical natural assets". For those reviewers who give their assent, we will be publishing their names alongside the published article.

Nature Ecology & Evolution offers a Transparent Peer Review option for new original research manuscripts submitted after December 1st, 2019. As part of this initiative, we encourage our authors to support increased transparency into the peer review process by agreeing to have the reviewer comments, author rebuttal letters, and editorial decision letters published as a Supplementary item. When you submit your final files please clearly state in your cover letter whether or not you would like to participate in this initiative. Please note that failure to state your preference will result in delays in accepting your manuscript for publication.

Cover suggestions

As you prepare your final files we encourage you to consider whether you have any images or illustrations that may be appropriate for use on the cover of Nature Ecology & Evolution.

If your image is selected, we may also use it on the journal website as a banner image, and may need

20to make artistic alterations to fit our journal style.

Nature Ecology & Evolution has now transitioned to a unified Rights Collection system which will allow our Author Services team to quickly and easily collect the rights and permissions required to publish your work. Approximately 10 days after your paper is formally accepted, you will receive an email in providing you with a link to complete the grant of rights. If your paper is eligible for Open Access, our Author Services team will also be in touch regarding any additional information that may be required to arrange payment for your article.

Please note that *Nature Ecology & Evolution* is a Transformative Journal (TJ). Authors may publish their research with us through the traditional subscription access route or make their paper immediately open access through payment of an article-processing charge (APC). Authors will not be required to make a final decision about access to their article until it has been accepted. [Find out more about Transformative Journals](https://www.springernature.com/gp/open-research/transformative-journals)

Authors may need to take specific actions to achieve [compliance with funder and institutional open access mandates](https://www.springernature.com/gp/open-research/funding/policy-compliance-faqs). If your research is supported by a funder that requires immediate open access (e.g. according to [Plan S principles](https://www.springernature.com/gp/open-research/plan-s-compliance)) then you should select the gold OA route, and we will direct you to the compliant route where possible. For authors selecting the subscription publication route, the journal's standard licensing terms will need to be accepted, including [a self-archiving-and-license-to-publish](https://www.nature.com/nature-portfolio/editorial-policies/self-archiving-and-license-to-publish). Those licensing terms will supersede any other terms that the author or any third party may assert apply to any version of the manuscript.

[REDACTED]

21[REDACTED]

Reviewer #1:

Remarks to the Author:

The authors have adequately addressed all my comments. The manuscript has been substantially improved in terms of organization, clarity, and scientific rigor. I have no doubt that this work is important and timely, and will be influential in years to come.

Jianguo (Jingle) Wu

Reviewer #3:

Remarks to the Author:

I have no further comments to make on this manuscript. I think the authors have sufficiently addressed my previous comments. I recommend this paper for publication.

Final Decision Letter:

13th October 2022

Dear Becky,

We are pleased to inform you that your Article entitled "Mapping the planet's critical natural assets", has now been accepted for publication in Nature Ecology & Evolution.

Over the next few weeks, your paper will be copyedited to ensure that it conforms to Nature Ecology and Evolution style. Once your paper is typeset, you will receive an email with a link to choose the appropriate publishing options for your paper and our Author Services team will be in touch regarding any additional information that may be required

Due to the importance of these deadlines, we ask you please us know now whether you will be difficult to contact over the next month. If this is the case, we ask you provide us with the contact information (email, phone and fax) of someone who will be able to check the proofs on your behalf, and who will be available to address any last-minute problems . Once your paper has been scheduled for online publication, the Nature press office will be in touch to confirm the details.

22Acceptance of your manuscript is conditional on all authors' agreement with our publication policies (see www.nature.com/authors/policies/index.html). In particular your manuscript must not be published elsewhere and there must be no announcement of the work to any media outlet until the publication date (the day on which it is uploaded onto our web site).

Please note that *Nature Ecology & Evolution* is a Transformative Journal (TJ). Authors may publish their research with us through the traditional subscription access route or make their paper immediately open access through payment of an article-processing charge (APC). Authors will not be required to make a final decision about access to their article until it has been accepted. [Find out more about Transformative Journals](https://www.springernature.com/gp/open-research/transformative-journals)

Authors may need to take specific actions to achieve [compliance with funder and institutional open access mandates](https://www.springernature.com/gp/open-research/funding/policy-compliance-faqs). If your research is supported by a funder that requires immediate open access (e.g. according to [Plan S principles](https://www.springernature.com/gp/open-research/plan-s-compliance)) then you should select the gold OA route, and we will direct you to the compliant route where possible. For authors selecting the subscription publication route, the journal's standard licensing terms will need to be accepted, including [those licensing terms](https://www.nature.com/nature-portfolio/editorial-policies/self-archiving-and-license-to-publish) will supersede any other terms that the author or any third party may assert apply to any version of the manuscript.

We welcome the submission of potential cover material (including a short caption of around 40 words) related to your manuscript; suggestions should be sent to Nature Ecology & Evolution as electronic files (the image should be 300 dpi at 210 x 297 mm in either TIFF or JPEG format). Please note that such pictures should be selected more for their aesthetic appeal than for their scientific content, and that colour images work better than black and white or grayscale images. Please do not try to design a cover with the Nature Ecology & Evolution logo etc., and please do not submit composites of images

23related to your work. I am sure you will understand that we cannot make any promise as to whether any of your suggestions might be selected for the cover of the journal.

You can generate the link yourself when you receive your article DOI by entering it here: <http://authors.springernature.com/share>.

[REDACTED]

P.S. Click on the following link if you would like to recommend Nature Ecology & Evolution to your librarian <http://www.nature.com/subscriptions/recommend.html#forms>

** Visit the Springer Nature Editorial and Publishing website at http://editorial-jobs.springernature.com?utm_source=ejP_NEcoE_email&utm_medium=ejP_NEcoE_email&utm_campaign=ejp_NEcoE for more information about our career opportunities. If you have any questions please click [here](mailto:editorial.publishing.jobs@springernature.com). **